# WebDART: Dynamic Decomposition and Re-planning for Complex Web Tasks

## Abstract

Large-language-model (LLM) agents are becoming competent at straightforward web tasks, such as opening an item page or submitting a form, but still struggle with objectives that require long-horizon navigation, large-scale information extraction, and reasoning under constraints. We present WebDART, a general framework that enables a single LLM to handle such complex chores. WebDART (i) *dynamically decomposes* each objective into three focused subtasks—navigation, information extraction, and execution—so the model concentrates on one skill at a time, and (ii) *continuously re-plans* the decomposition as new webpages are revealed, taking advantage of newly discovered filters or shortcuts and avoiding redundant exploration. Evaluated on WebChoreArena, WebDART lifts end-to-end success rates by up to 13.7 percentage points over previous state-of-the-art agents, while matching their performance on the easier WebArena suite and completing tasks with up to 14.7 fewer navigation steps. Code will be publicly available.

## 1 Introduction

LLM-powered web agents have recently shown promising abilities in web navigation tasks (Drouin et al., 2024; He et al., 2024; Wei et al., 2025; Yang et al., 2024a; Pan et al., 2024; Song et al., 2024). Benchmarks such as WebArena (Zhou et al., 2023) demonstrate that these agents achieve reasonable accuracy on simple objectives, highlighting their potential as general-purpose automation tools. However, when the objectives require more complex reasoning and multi-step exploration, the performance of these agents often collapses. As shown in Figure 1, on WebChoreArena (Miyai et al., 2025), a benchmark designed to test higher-complexity web tasks, agents powered by GPT-4o achieve only 8.0% accuracy on tasks across different web domains, far below the 46.6% accuracy on WebArena. This gap highlights a critical weakness of current worflows: while sufficient for simple goals, they are not well equipped for tasks demand multi-step reasoning, long-horizon navigation, and structured information processing.

A closer examination reveals that the difficulty arises from cognitive overload. Complex tasks require agents to simultaneously navigate across multiple web pages, extract and track large amounts of information, and reason under constraints. Consider the following task from WebChore-Arena (Miyai et al., 2025): *"Tell me the top 3 products with the highest number of reviews in Home Audio of Electronics within the price range of $1,000 to $9,999"*. As illustrated in Figure 1, product information is distributed across multiple nested web pages. Each page may contain tens of products with attributes such as price and number of reviews. To complete this objective, current LLM agents (Yang et al., 2024a; Chezelles et al., 2024) attempt to tackle all these aspects in a single process: while browsing through pages, they must also keep track of which products meet the price requirement, remember which ones they have already seen, and simultaneously apply the logic needed to determine the top three by number of reviews. This often overwhelms the agent, leading to frequent mistakes such as missing relevant information, forgetting the user instructions, and incorrect analysis (Miyai et al., 2025).

In contrast, human experts may naturally break the task into distinct steps: ❶ first narrowing down to the pages within the desired price range, ❷ then collecting and recording the attributes of candidate products, and ❸ finally ranking the products by number of reviews. This stepwise approach reduces

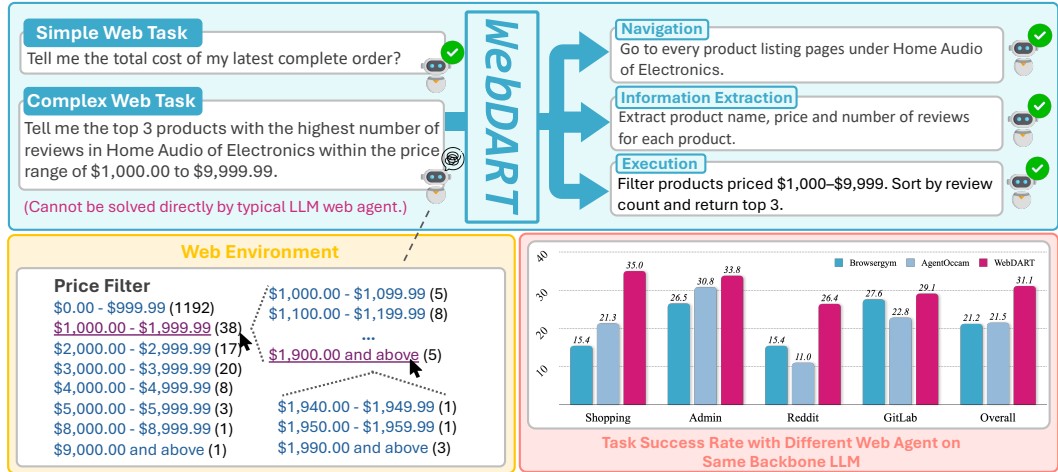

Figure 1: (Top) Existing LLM-based web agents perform well on simple tasks, but their success rates drop on complex tasks that require non-trivial reasoning, such as applying a price-range filter (bottom left).WEBDART overcomes this limitation by dynamically decomposing the objective into three subtasks: navigation, information extraction, and execution. (Bottom right) Consequently,WEBDART significantly outperforms the current state of the art on WebChoreArena across all task categories. Backbone LLM: GPT-5.

complexity of the task and makes the problem tractable, whereas forcing all operations to occur simultaneously overwhelms current agents and leads to frequent errors.

Motivated by this, we propose **WEBDART** (**D**ecomposition & **A**daptive **R**e-planning for **T**asks), a framework that adaptively decomposes complex web tasks into simpler, modular subtasks. Unlike the typical agentic flow, where navigation, information extraction, and execution are interleaved in a single process, WEBDART separates the original complex tasks into these three subtasks. We adopt these three subtasks because complex web tasks typically require distinct agent abilities: browsing through multiple pages, extracting relevant information, and performing analysis or acting on the results. One example of the decomposition is shown in Figure 1, where we leverage the LLM to generate a decomposition conditioned on both the task description and the initial web environment. The task decomposition reduces the cognitive burden on the LLM and makes complex objectives more tractable by allowing the agent to focus on one subtask at a time.

However, an initial decomposition based only on the task description may be suboptimal. There are multiple ways to decide what information should be collected during navigation versus deferred to later analysis, and these trade-offs cannot always be known in advance. Moreover, as the agent explores, new web elements such as filters or sort options may appear that were unavailable at the beginning but can drastically reduce navigation effort. For example, in Figure 1, the initial navigation subtask is specified as *"visit every product listing page under Home Audio of Electronics"*. Once the agent enters the product page, it may discover a price filter that allows it to restrict results to $1,000 to $9,999 and avoid traversing irrelevant pages. To exploit such opportunities, WEBDART incorporates a *dynamic replanning* mechanism during navigation that allows the agent to revise its plan after each step based on newly observed pages. This adaptive adjustment helps correct mistakes and eliminates redundant exploration. Together, task-adaptive decomposition and navigation replanning enable WEBDART to achieve higher accuracy with lower cost.

We perform extensive evaluation of our method on both WebChoreArena and WebArena across three different LLM backbones. With the proposed decomposition framework, WEBDART improves state-of-the-art agent frameworks including BrowserGym (Chezelles et al., 2024) and AgentOccam (Yang et al., 2024a) by up to 13.7% on the complex tasks in WebChoreArena. Our method also achieves similar performance on WebArena compared to existing state-of-the-arts, demonstrating its robustness and flexibility. Finally, by combining the dynamic re-planning module, the accuracy of our method can be further increased by 7.7% on the shopping tasks in WebChoreArena while reducing the average navigation steps by 14.7.

## 2 RELATED WORK

**Simulated web-agent environments.** Progress on web agents has largely mirrored progress on the testbeds available to them. The first generation of benchmarks—MiniWoB and MiniWoB++ (Liu et al., 2018)—offers canvas-rendered "toy" sites that evaluate low-level actions such as clicking or typing within a single, synthetic page. WebShop keeps the single-domain setting but increases realism by simulating a full e-commerce catalogue, requiring agents to search, filter, and purchase items.

The next wave introduces multi-domain, fully functional sites. WebArena (Zhou et al., 2023) hosts independent applications for shopping, forums, software development, and content management, thereby capturing a broader range of real-world behaviours. More recent suites push two frontiers. (1) Multimodality: VisualWebArena (Koh et al., 2024) and WebVoyager (He et al., 2024) add image inputs so that agents must reason jointly over text and vision. (2) Task complexity: WebChore-Arena (Miyai et al., 2025) reuses the WebArena sites but issues longer "chores" that demand capabilities beyond ordinary browsing—e.g., arithmetic, cross-page memory, and long-horizon planning.

Our study targets the text-only setting and therefore evaluates on WebArena and WebChoreArena, which together provide diverse domains and richly composed task intents while remaining fully reproducible.

**LLM-powered web agents.** Current web agents can be grouped into three broad lines of work. (1) Leveraging execution feedback. Prompting schemes such as ReAct and its derivatives let an LLM interleave reasoning and actions during a rollout (Yao et al., 2023; Mialon et al., 2023; Hong et al., 2024; Yang et al., 2024b; Amayuelas et al., 2025; Yang et al., 2025). Subsequent methods reuse the generated trajectories to refine future attempts: AWM distils frequently successful action patterns (Wang et al., 2024); Auto Eval & Refine trains an external evaluator and invokes self-reflection (Pan et al., 2024; Shinn et al., 2023); WebPilot explores alternate paths with an MCTS-style search (Zhang et al., 2025b). (2) Synthesising auxiliary data. Learn-by-Interact creates synthetic tasks, relabels the resulting trajectories with hindsight (Su et al., 2025; Li et al., 2020), and retrieves them at inference time, while AgentSymbiotic uses a large–small model pair to co-generate training examples (Zhang et al., 2025a). These approaches boost accuracy when the synthetic tasks closely match the evaluation set but risk data contamination and often degrade when distributions diverge. (3) Optimising the interface. AgentOccam shows that simply pruning the DOM observation and restricting the action set already yields large gains and is now a common preprocessing step (Yang et al., 2024a). (4) Finetuned web agents represent another important line of work complementary to training-free designs like ours. These approaches explicitly fine-tune an LLM policy using domain-specific trajectories to encode stronger priors for multi-step decision making. Recent examples include curriculum-based reinforcement learning agents that evolve their own training distribution over time (Qi et al., 2024), models that learn webpage-specific contextualization layers to filter DOM observations before acting (Lee et al., 2025), and GUI-generalist agents, finetuned on large multimodal UI demonstrations, to perform precise manipulation and element grounding (Qin et al.). While fine-tuning often yields higher in-distribution accuracy, these methods typically require expensive data generation and can be brittle under distribution shifts. In contrast, our approach instead relies on structured task decomposition and interface optimization to achieve strong generalization without additional training cost.

WEBDART departs from all of the above. (i) It is *training-free*: no extra rollouts, synthetic data, or fine-tuning are required. (ii) It tackles long-horizon chores through *dynamic task decomposition*: during execution, the agent continually observes the current webpage and adaptively refines a three-part plan—navigation, information extraction, and execution—allowing the same frozen backbone LLM to focus on one capability at a time. This simple yet principled design delivers state-of-the-art results on both WebArena and WebChoreArena.

## 3 METHOD

In this paper, we focus on *text-based* web agents, although the proposed approach naturally extends to multimodal environments. Each task is specified by a natural-language instruction and a ground-

Figure 2: **Overview of the WEBDART framework.** A complex web task is dynamically decomposed into three sequential subtasks. (1) **Navigation:** the agent explores the site—issuing actions such as `click`, `type`, and `go_back`—to gather every page that could contain the required information. (2) **Information extraction:** given these pages, a dedicated module isolates task-relevant content and converts it into a standardised, structured form based on the objective. (3) **Execution:** the extracted data are analysed to meet the task constraints, e.g., by generating and running Python code on the fly to perform filtering, aggregation, or other computations.

truth target for evaluation. The agent receives the instruction and interacts with a web environment whose pages are represented as accessibility trees, aiming to fulfil the stated objective.

Figure 2 illustrates the WEBDART workflow. A complex web task is first *dynamically decomposed* into a sequence of modular subtasks that are executed in order. The central challenge is to choose a decomposition whose subtasks are both tractable and complementary.

Empirically, most web tasks require three distinct capabilities:

1. **Navigation**: browsing across multiple pages to locate candidate information;
2. **Information extraction**: converting raw page content into structured records;
3. **Execution**: analysing the collected data or acting on the results.

Guided by this observation, WEBDART decomposes every complex task into the ordered subtasks of *navigation*, *information extraction*, and *execution*, continually updating intermediate objectives as new observations arrive. In what follows, we first describe the decomposition strategy (Section 3.1), and then detail the navigation (Section 3.2), information-extraction (Section 3.3), and execution (Section 3.4) modules.

## 3.1 TASK DECOMPOSITION

A web task can be decomposed in several ways, and the most suitable granularity depends on the structure of the target site. Consider the task in Figure 2: *"Calculate the total number of comments on the 30 most-commented posts in the OldSchoolCool forum."* Two natural decompositions are

- **Tightly coupled.** Embed the numeric constraint in the navigation objective: *"Browse Old-SchoolCool and open the 30 most-commented posts."*
- **Conservative.** Keep navigation agnostic to the constraint: *"Browse OldSchoolCool and visit all post-listing pages."* Identifying the top 30 posts is then left to the analysis stage.

Both options are valid, but their efficiency hinges on site features. If the forum provides a `Sort by: most commented` control, the tight plan is ideal—it satisfies the constraint while touching

only a handful of pages. Conversely, when such affordances are absent (or the total number of pages is already small), the conservative plan is simpler and more reliable: the agent just collects every listing page and defers heavy reasoning to later stages.

Because these interface aids are unknown *a priori*, WEBDART adopts the conservative scheme by default and adapts opportunistically. Specifically, all data–centric operations—filtering, sorting, ranking—are initially assigned to execution, while navigation is limited to page discovery. To steer the LLM toward this partitioning, the prompt $\mathbf{p}$ contains three in-context examples that consistently push constraint handling to later stages:

$$f : (\mathcal{T}, \mathbf{p}) \longrightarrow (\mathcal{T}_{\text{nav}}, \mathcal{T}_{\text{ie}}, \mathcal{T}_{\text{exec}}),$$

where $f(\cdot)$ is the LLM and the outputs $\mathcal{T}_{\text{nav}}, \mathcal{T}_{\text{ie}}, \mathcal{T}_{\text{exec}}$ are the navigation, information-extraction, and execution objectives.

During navigation the agent may encounter helpful widgets (*e.g.*, the aforementioned sort button) that can fulfill part of the constraint immediately. When detected, WEBDART invokes *dynamic re-planning*: the current navigation goal $\mathcal{T}_{\text{nav}}$ is updated on-the-fly, allowing the agent to skip irrelevant pages and accelerate completion. Details of this mechanism are presented in Section 3.2.

**Fast-path routing.** Finally, the decomposition module also incorporates a lightweight router that decides whether the task can be satisfied with only a *subset* of the three modules. For instance, the instruction "Post `"Hello, world!"` on `/OldSchoolCool`" requires navigation (and possibly execution) but no information extraction; the router therefore bypasses the extraction stage and invokes the minimal workflow.

## 3.2 NAVIGATION

The navigation module drives the agent through the website, issuing low-level browser actions until every page that might contain task-relevant information has been visited. Our interactive setup follows prior work (Yang et al., 2024a; Wang et al., 2024; Zhang et al., 2025a).

At time step $t$ the agent outputs a pair $(r_t, a_t)$: a natural-language reasoning trace $r_t$ and an action $a_t \in \mathcal{A}$, where $\mathcal{A} = \{\text{click}, \text{type}, \text{go\_back}, \text{stop}\}$. The choice is conditioned on (i) the current navigation objective $\mathcal{T}_{\text{nav}}$, (ii) the current observation $o_t$ (the page rendered as an accessibility tree), and (iii) the interaction history $\mathbf{h}_t = (\mathbf{o}_{1:t-1}, \mathbf{a}_{1:t-1}, \mathbf{r}_{1:t-1})$. After execution, $(r_t, a_t)$ is appended to the history; when the agent finally emits `stop` at step $T$, the full interaction history $\mathbf{h}_T = (\mathbf{o}_{[1:T]}, \mathbf{a}_{[1:T]}, \mathbf{r}_{[1:T]})$ is passed to the information-extraction module. Figure 3 illustrates the workflow.

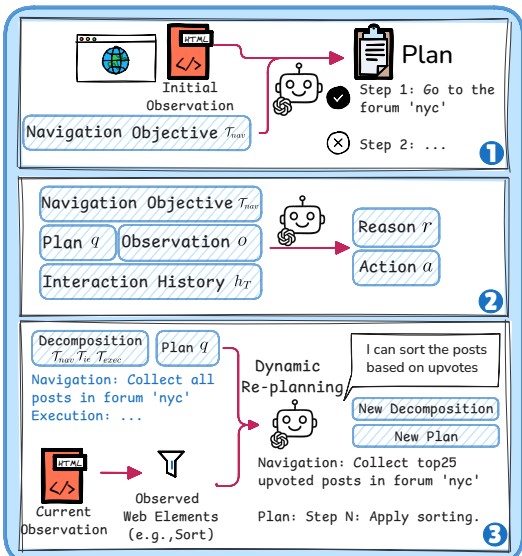

Figure 3: Illustration of the WEBDART framework in navigation. An initial plan is generated before starting navigation. The navigation agent issues an action at each step. When new web elements (e.g., filters, sorting options) appear, the dynamic re-planning module updates the decomposition and plan, enabling the agent to adapt its strategy for more efficient execution.

**Plan-guided browsing.** Before the first action, the LLM is given the navigation objective $\mathcal{T}_{\text{nav}}$ and the initial page $o_0$ and asked to generate a high-level plan $\mathbf{q}_0$. The plan lists (i) pages to visit, (ii) information to capture, and (iii) a stopping criterion. During browsing the agent is prompted with $\mathcal{T}_{\text{nav}}$, the current plan $\mathbf{q}_{t-1}$, the observation $o_t$, and the history $\mathbf{h}_t$. Conditioning on $\mathbf{q}_{t-1}$ stabilises behaviour and substantially reduces premature termination (sample plans are shown in Appendix A.1.2).

**Dynamic replanning.** The conservative decomposition from Section 3.1 defers all constraint handling to the execution stage; this guarantees coverage but can be wasteful when helpful interface widgets (filters, sort menus, etc.) appear mid-navigation. To exploit such shortcuts, the agent performs *dynamic replanning*.

At the start of each step $t$ the agent evaluates, based on $(o_t, \mathbf{h}_{t-1}, \mathbf{q}_{t-1}, \mathcal{T})$, whether the navigation objective or plan should be revised. If a useful widget has been discovered, it outputs an updated pair $(\mathcal{T}_{\text{nav}}^t, \mathbf{q}_t)$ that incorporates the shortcut; otherwise it keeps the previous version. The (possibly) updated objective and plan are fed back into the action-selection prompt to produce $(r_t, a_t)$.

Dynamic replanning preserves the safety of a conservative start while allowing the agent to exploit opportunistic efficiencies—for example, switching from "visit every listing page" to "apply `sort by: most-commented` and scan only the first 30 posts." The prompt template used for this mechanism is provided in Appendix A.1.5.

### 3.3 Information Extraction

When navigation ends at step $T$, we obtain the transcript $\mathbf{h}_T = (\mathbf{o}_{1:T}, \mathbf{a}_{1:T}, \mathbf{r}_{1:T})$, where $\mathbf{o}_{1:T}$ contains every page the agent observed. Blindly extracting from *all* pages would add substantial noise—for example, products in the wrong category or outside a specified price range. The extraction module therefore proceeds in two stages:

**Page selection.** An LLM is given the original task $\mathcal{T}$, the navigation objective $\mathcal{T}_{\text{nav}}$, and the full history $\mathbf{h}_T$. It returns an index set $\mathcal{I} \subseteq \{1, \ldots, T\}$ that marks the pages most likely to contain the required information (prompt template in Appendix A.1.3).

**Field extraction.** For each chosen page $o_t$ $(t \in \mathcal{I})$, a second LLM call extracts the target fields—*e.g.*, post title and comment count—directly from the page's accessibility tree, producing a uniform JSONL record. The resulting structured collection is passed to the execution module.

We also experimented with an *LLM-generated parser* baseline, where the model generates code on the fly to traverse the accessibility tree of each $o_t$. In practice, this approach proved brittle: accessibility trees are deeply nested and site-specific, and minor layout changes frequently break the generated code. Prompt-based extraction avoids these issues and requires no hand-crafted logic; therefore, WEBWEAVER adopts it as the default strategy.

### 3.4 Execution

The execution module converts the structured records produced by the information-extraction stage into the final deliverable requested by the task. Depending on $\mathcal{T}_{\text{exec}}$, this entails one of two subroutines.

**Data-analysis objectives.** When the task calls for statistics, rankings, or other derived quantities, the agent generates and runs code (Python by default) over the extracted JSON file. Typical operations include filtering under constraints, aggregation, and sorting. To increase robustness we adopt a *self-reflection* loop (Shinn et al., 2023): if the program throws an exception, the LLM examines the traceback, amends the code, and re-executes it until success or a timeout. Implementation details are provided in Appendix A.2.

**Action-oriented objectives.** Some tasks require injecting the computed result back into the environment—for example, posting a summary to a forum or submitting a completed form. In these cases the module invokes a short-horizon navigation policy that is initialised with the analysis output (e.g., the text to post or the value to enter). Because the destination elements are already known, this policy is far simpler and more reliable than the primary navigation module, yet it preserves the same interface and action space.

In both settings, once the required code or interactions have concluded, the agent returns the task's final answer and the execution stage terminates.

Table 1: Results on the **WebChoreArena** benchmark across different web domains (Shopping, Reddit, Admin, GitLab). WEBDART consistently outperforms all baselines across models , achieving the highest overall success rate. Results with † are reported by WebChoreArena (Miyai et al., 2025).

| Model | Method | Shopping | Reddit | Admin | GitLab | Overall |
|---|---|---|---|---|---|---|
| GPT-5 | SteP (Sodhi et al., 2023) | 2.6 | 4.4 | 0.7 | 4.7 | 3.1 |
| | BrowserGym (Chezelles et al., 2024) | 15.4 | 15.4 | 26.5 | 27.6 | 21.2 |
| | AWM (Wang et al., 2024) | 18.0 | 14.3 | 30.3 | 26.8 | 22.4 |
| | AgentOccam (Yang et al., 2024a) | 21.3 | 11.0 | 30.8 | 22.8 | 21.5 |
| | WEBDART | **35.0**$_{\uparrow13.7}$ | **26.4**$_{\uparrow10.0}$ | **33.8**$_{\uparrow3.0}$ | **29.1**$_{\uparrow1.5}$ | **31.1**$_{\uparrow8.7}$ |
| GPT-4o | SteP (Sodhi et al., 2023) | 2.6 | 0.0 | 0.0 | 4.7 | 1.8 |
| | BrowserGym† (Chezelles et al., 2024) | 0.9 | 5.5 | 2.3 | 3.9 | 3.2 |
| | AWM (Wang et al., 2024) | 3.4 | 8.8 | 4.5 | 4.7 | 5.4 |
| | AgentOccam† (Yang et al., 2024a) | 10.3 | 9.9 | 4.5 | 7.1 | 8.0 |
| | WEBDART | **18.8**$_{\uparrow8.5}$ | **19.8**$_{\uparrow9.9}$ | **12.9**$_{\uparrow8.4}$ | **9.4**$_{\uparrow2.3}$ | **15.2**$_{\uparrow7.2}$ |
| GLM-4.5-air-fp8 | SteP (Sodhi et al., 2023) | 0.0 | 2.2 | 1.5 | 2.4 | 1.5 |
| | BrowserGym (Chezelles et al., 2024) | 6.0 | 4.8 | 6.1 | 9.4 | 6.6 |
| | AWM (Wang et al., 2024) | 0.9 | 5.6 | 4.3 | 8.7 | 4.9 |
| | AgentOccam (Yang et al., 2024a) | 18.8 | 4.4 | 11.4 | 8.7 | 10.8 |
| | WEBDART | **26.5**$_{\uparrow7.7}$ | **16.5**$_{\uparrow10.9}$ | **18.9**$_{\uparrow7.5}$ | **15.4**$_{\uparrow6.0}$ | **19.3**$_{\uparrow8.5}$ |

# 4 EXPERIMENT RESULTS AND ANALYSIS

## 4.1 EXPERIMENT SETUP

**Environment.** We conduct experiments on two benchmarks: **WebChoreArena** and **WebArena**. WebChoreArena (Miyai et al., 2025) is our primary evaluation benchmark, as it extends the WebArena (Zhou et al., 2023) environment with more realistic and challenging chores that require handling constraints, information extraction, and data analysis in addition to navigation. These tasks better reflect the complexity of real-world web usage and thus serve as the main testbed for demonstrating the effectiveness of our method. In parallel, we also evaluate on WebArena tasks to ensure that our approach does not reduce performance on simpler navigation-oriented objectives. Both benchmarks share the same set of interactive web environments (e.g., shopping, administration, forums, and code management), which allows us to make a direct comparison between simple and complex tasks under consistent conditions.

**Baselines.** We compare WEBDART against four baselines: **SteP** (Sodhi et al., 2023), **BrowserGym** (Chezelles et al., 2024), **AWM** (Wang et al., 2024) and **AgentOccam**. SteP (Sodhi et al., 2023) (Stacked LLM Policies) is a method that decomposes the web-agent policy space into multiple subpolicies, dynamically composing them to adapt to task complexity. BrowserGym (Chezelles et al., 2024) provides a unified evaluation framework for web agents with standardized observation and action spaces, enabling fair and reproducible comparisons across different benchmarks. AWM (Wang et al., 2024) induce commonly reused routines from web tasks to guide subsequent generations. AgentOccam (Yang et al., 2024a) is our main baseline, as it employs a navigation agent design closely aligned with ours; by focusing on observation and action spaces that match LLM pretraining distributions, it achieves strong results on WebArena without relying on in-context examples or external search. Together, these baselines allow us to evaluate WEBDART against diverse approaches while ensuring a fair comparison with a closely related navigation agent. We compare WEBDART with these baselines with three different backbone LLMs including GPT-5, GPT-4o, and GLM-4.5-air-fp8. The configurations for each model and experiment setup is detailed in Appendix A.2

## 4.2 EVALUATION ON COMPLEX WEB TASKS.

Table 1 presents the main results on the **WebChoreArena** benchmark, which evaluates agent performance on complex multi-step web tasks involving constraints and information extraction. We compare WEBDART against three baselines: SteP, AWM, BrowserGym, and AgentOccam, under three different backbone models (GPT-5, GPT-4o, and GLM-4.5-air-fp8).

Table 2: Efficiency evaluation of **dynamic re-planning** on WebChoreArena with GPT-4o as backbone LLM. We report accuracy and average navigation steps.

| | Shopping | | Reddit | | Admin | | GitLab | |
|---|---|---|---|---|---|---|---|---|
| | Accuracy | Avg. Steps | Accuracy | Avg. Steps | Accuracy | Avg. Steps | Accuracy | Avg. Steps |
| WEBDART | 18.8 | 32.9 | 19.8 | 25.1 | 12.9 | 16.7 | 9.4 | 23.3 |
| + Dynamic Re-planning. | $26.5_{\uparrow 7.7}$ | $18.2_{\downarrow 14.7}$ | $20.9_{\uparrow 1.1}$ | $21.1_{\downarrow 4.0}$ | $13.6_{\uparrow 0.7}$ | $17.7_{\uparrow 1.0}$ | $11.1_{\uparrow 1.7}$ | $21.2_{\downarrow 2.1}$ |

Across all model backbones, WEBDART achieves the highest overall success rates, demonstrating its robustness and effectiveness on complex tasks. With GPT-5, WEBDART reaches 31.1 overall, outperforming SteP (3.1), BrowserGym (21.2), AWM (22.4), and AgentOccam (21.5). The gains are particularly pronounced in the Shopping and Reddit domains, where WEBDART improves over AgentOccam by +13.7 and +15.4 points respectively. This highlights the advantage of shifting constraint handling to the data analysis stage, which reduces error propagation from fragile navigation.

The improvements are consistent for GPT-4o, where WEBDART achieves 15.2 overall compared to 8.0 for AgentOccam, and for GLM-4.5-air-fp8, where WEBDART reaches 19.3 overall compared to 10.8 for AgentOccam. These results suggest that our method generalizes across different backbone models, even when the underlying LLM has weaker navigation or reasoning capabilities.

We also note that SteP underperforms significantly on WebChoreArena compared to other baselines and WEBDART, reflecting its limited ability to handle tasks with deep constraint hierarchies. In contrast, WEBDART consistently maintains a strong margin over all baselines, confirming that decomposition is the key to solving complex web chores efficiently.

### 4.3 EVALUATION OF DYNAMIC RE-PLANNING.

In Section 3.2, we introduced *dynamic re-planning*, where the navigation agent adapts its decomposed subtasks and plan based on newly discovered web elements (e.g., filters or sorting options) that can directly apply task constraints. This mechanism aims to reduce redundant navigation and improve efficiency, while preserving or even improving accuracy. Table 2 reports the results of comparing agents with and without dynamic re-planning across four domains in using GPT-4o as the backbone model. We report both task accuracy and the average number of navigation steps.

The results show that dynamic re-planning substantially reduces the number of navigation steps. In the Shopping domain, the average navigation steps decrease from 32.9 to 18.2 while accuracy improves from 18.8% to 26.5%. A similar trend is observed in Reddit, where the step count drops from 25.1 to 20.8, with a modest accuracy gain (19.8% to 20.9%). The only exception occurs in the Shopping Admin domain. This is because the website inherently relies on numerous filters and sorting elements, without which the tasks cannot be completed. These improvements confirm that dynamically adapting the decomposition and plan allows the agent to bypass unnecessary exploration and focus on relevant parts of the environment.

Table 3: Results on the **WebArena** benchmark. Bold numbers indicate the best performance, and underlined numbers indicate the second best. All the methods are tested using GPT-4o as backbone model. The baseline results are taken from previous works (Zhang et al., 2025b; Song et al., 2024).

| Method | Shopping | Admin | Reddit | GitLab | Overall |
|---|---|---|---|---|---|
| WebArena (Zhou et al., 2023) | 13.9 | 10.4 | 6.6 | 15.0 | 11.5 |
| AutoEval (Pan et al., 2024) | **39.6** | 20.9 | 20.8 | 25.0 | 26.6 |
| AWM (Wang et al., 2024) | 32.1 | 29.1 | 54.7 | 35.0 | 37.7 |
| SteP (Sodhi et al., 2023) | 36.9 | 24.2 | 59.4 | 31.7 | 38.0 |
| HybridAgent (Song et al., 2024) | 25.7 | 41.2 | 51.9 | 44.4 | 40.8 |
| WebPilot (Zhang et al., 2025b) | 36.9 | 24.7 | 65.1 | 39.4 | 41.5 |
| AgentOccam (Yang et al., 2024a) | 37.4 | **44.0** | 66.0 | 38.9 | 46.6 |
| WEBDART | 36.0 | 41.2 | **67.9** | **47.2** | **48.1** |

Overall, these results validate the effectiveness of dynamic re-planning as a complementary strategy in WEBDART. By allowing the agent to adjust its task structure in real time, we achieve shorter navigation paths and, in several domains, notable accuracy improvements.

### 4.4 EVALUATION ON SIMPLE NAVIGATION TASKS.

While WEBDART is primarily designed for complex web tasks involving constraints and analysis, it is also important to verify that the framework does not degrade performance on simpler navigation-oriented tasks. To this end, we evaluate on the original **WebArena** benchmark, where most tasks can be completed through direct navigation without requiring decomposition. For these tasks, we adjust the agent to bypass the decomposition stage and focus solely on the navigation module.

Table 3 reports the results, comparing WEBDART against a wide range of existing web agents. We observe that WEBDART achieves competitive or superior performance across domains, reaching an overall success rate of 48.1, which is higher than all baselines including AgentOccam (46.6).

These results confirm that WEBDART maintains robustness across task types: it significantly improves over baselines in complex settings by leveraging decomposition, while also remaining competitive on simpler navigation tasks by bypassing unnecessary modules. This adaptability demonstrates the generality of our design.

### 4.5 CASE STUDY.

We further present case study to visualize how dynamic re-planning enhances WEBDART in Table 4. In the first example, the agent initially plans to traverse every page in a product category, but upon detecting a drop-down menu that adjusts the number of displayed products, the plan is revised to greatly reduce navigation steps. This shows how re-planning exploits newly discovered web elements to improve efficiency. In the second case, the agent's initial decomposition requires visiting all forums to collect a user's submissions, which is infeasible. Once it identifies that the user profile page already lists submissions with a direct link, the plan and the navigation objective is updated to extract information more directly, correcting a flawed decomposition. Finally, in the third case, the agent relies on keyword search that produces irrelevant results. Dynamic re-planning detects the mismatch and redirects the strategy to the actual forum page, enabling the agent to recover from misleading navigation. Together, these examples demonstrate that dynamic re-planning allows the agent to correct initial mistakes and maintain robustness in complex web environments.

Table 4: Case studies of dynamic re-planning in WEBDART.

| Original Task | Initial Navigation Objective | Web Elements (Description) | Navigation Objective after replanning |
|---|---|---|---|
| Calculate average product price in *Diet & Sports Nutrition* | Plan includes navigating to *Diet & Sports Nutrition* category and going over all the pages. | Menu to select number of products displayed in each page. | Add the step changing the number of products displayed each page from 12 to 36. |
| Count submissions by specific user *thebelsnickle1991* in each forum | Decomposition requires traversing submissions in every forum alphabetically, leading to endless exploration. | Button to submission listing page under the user profile page. | Revise plan to extract directly from the profile page and aggregate submissions. |
| Count unique users among top 600 hottest submissions in *nyc* forum | Initial plan relies on keyword search for "nyc," which returns unrelated articles. | Direct link to the *nyc* forum and its sorting options. | Bypass search results and directly navigate to the forum page before collecting data. |

## 5 CONCLUSION

We introduced WEBDART, a framework that enhances web agents on complex tasks through explicit subtask decoupling and dynamic re-planning. By shifting constraint handling and other data-

related operations from navigation to the analysis stage, WEBDART reduces error propagation and alleviates the burden on fragile navigation processes. At the same time, dynamic re-planning enables the agent to adapt plans in real time when new web elements are discovered or when the initial decomposition is suboptimal. Experiments on WebChoreArena demonstrate that WEBDART improves task success rates by up to 13.7% over strong baselines while also reducing navigation steps, and evaluation on WebArena confirms that our method maintains performance on simpler tasks. Case studies further show how re-planning allows the agent to exploit new opportunities, correct inefficient strategies, and recover from misleading navigation paths, leading to more efficient and robust web automation.

## REPRODUCIBILITY STATEMENT

We have taken several steps to ensure the reproducibility of our work. The benchmarks used in our experiments, including WebChoreArena and WebArena, are publicly available and described in detail in Section 4.1. The implementation details of WEBDART, including decomposition and dynamic re-planning, are provided in Section 3, and additional examples and prompts are included in the Appendix A. Finally, we provide our source code as part of the supplementary materials.

## ETHICS STATEMENT

We have carefully reviewed the ICLR Code of Ethics and found no potential ethical issues related to our work. Our study does not involve human subjects, sensitive data, or applications that pose foreseeable risks of harm.

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

## A APPENDIX

### A.1 AGENT PROMPTS & EXAMPLES

Inside this section, we displayed the prompts as well as some intermediate outputs as demonstration examples for for each module of WEBDART.

#### A.1.1 DECOMPOSITION

The following prompt illustrates an example of decomposition for data-analysis objectives. It explicitly encourages a conservative strategy, as discussed in our method section, by deferring data-related operations to the analysis stage. In addition, we provide three in-context examples to help the LLM better follow this decomposition approach.

---

**Prompt - Decomposition**

You are conducting a complex web task that requires information from the web to answer correctly. Directly navigating the web environment to provide a final answer cannot always yield the correct result. Therefore, you need to decompose the task into two decoupled parts to complete it successfully.

The two parts are the navigation part and the analysis part. The navigation part involves visiting all pages that contain the data needed to solve the task. The observation, the accessibility tree of full web page, at each step will be recorded during navigation.
The analysis part involves extracting information from the observations and writing code to provide the final answer. Note that the extracted information processed during analysis part may be imperfect, which means they may include unnecessary data or not in correct format, you need to make sure the analysis code can be robust to handle such cases.

Another important consideration is to simplify the navigation, as it is a more challenging task. Ignore constraints such as ranges or filters in the navigation objective. Instead, include such constraints in the analysis part to be handled later.

Given the original complex user task and some tips for using the target website, decompose it into these two parts following this approach. Your output must follow this format with exact the same headers:

**### Part 1 – Navigation**

**### Part 2 – Analysis**

In addition, below are some decomposition examples for your reference:

**Example 1:**

User task "List the average rating for every movie genre, using only titles released between 2015 and 2024. Output: 'Drama : 8.1, Comedy : 7.4, . . .'"

### Part 1 – Navigation Go to the pages which include each film's genre, release year, and numeric user rating. Do not go to each film detail page if all the information is available in film listing page.

### Part 2 – Analysis Filter and only keep only films released 2015-2024. Compute the average rating per genre and show them as 'Drama : X.X, Comedy : Y.Y, . . .'.

**Example 2:**

User task "Among products tagged 'wireless earbuds', count how many cost below $50, $50-$99, and $100+. Return: '<50 : __, 50-99 : __, 100+ : __'."

---

### Part 1 – Navigation Visit the pages containing product title and price information for "wireless earbuds" products. Do not go to each product detail page if all the information is available in product listing page.

### Part 2 – Analysis Group the collected items by price brackets < $50, $50-$99, $100+. Count how many fall into each bracket and output the counts in the following format: '<50 : __, 50-99 : __, 100+ : __'

**Example 3:**

User task "In the travel forum, among the 200 latest hotel reviews, how many mention 'noise' or 'quiet' in the text? Give two numbers: noisy_count, quiet_count."

### Part 1 – Navigation Navigate to the pages including the text body of the hotel reviews in most recent order in the travel forum. Go over all hotel reviews in total. Do not go to each review detail page if all the information is available in review listing page.

### Part 2 – Analysis Only keep first 200 reviews. Search each saved review for the words "noise", "noisy" (noisy_count) and "quiet". Return two integers: noisy_count and quiet_count.

Below is one decomposition example generated conditioned on the prompt above:

---

**Example - Decomposition**

**Original Task:**

Extract the title of reviews with a rating of 2 or below out of 5 stars from 'Tea Gift Set for Tea Lovers - Includes Double Insulated Tea Cup 12 Uniquely Blended Teas and All Natural Honey Straws — Tea Gift Sets for Women Men — Tea Gifts Bag Presented in Beautiful Gift Bag' and output them as a list in alphabetical order, separeted by line breaks.

**Navigation Objective:**

Navigate to the product page for 'Tea Gift Set for Tea Lovers - Includes Double Insulated Tea Cup 12 Uniquely Blended Teas and All Natural Honey Straws — Tea Gift Sets for Women Men — Tea Gifts Bag Presented in Beautiful Gift Bag'. Visit the reviews section of the product and collect the review titles along with their star ratings.

**Analysis Objective:**

Filter the collected reviews to include only those with a rating of 2 stars or below. Extract the titles of these reviews and sort them in alphabetical order. Output the sorted titles as a list, with each title separated by a line break.

---

### A.1.2 NAVIGATION

In this section, we display the prompts for each part of navigation module and provide corresponding examples.

---

**Prompt - Navigation**

You are an AI assistant performing navigation tasks on a web browser. You will be provided with task objective, current step, web page observations, current plan, and interaction history. You need to issue an action for this step.

Your task is mainly about navigating to each page that may contain the needed information.

---

Generate the response in the following format: {output_specifications}

You are ONLY allowed to use the following action commands. Strictly adheres to the given format. Only issue one single action. {navigation_specifications}

{website_tips}

**Prompt - Navigation Plan Generation**

You are an AI assistant that generates initial plans for web navigation tasks. Given a task objective and an initial web page observation in accessibility tree, you need to create a clear, step-by-step plan that will guide the navigation agent.

Directly output the navigation plan in your response without other irrelevant information.

Your plan should be:

1. Clear and actionable

2. Broken down into logical steps

3. Specific enough to guide navigation

4. Concise, only including necessary steps. Do not dive into more pages if the current page already contains the needed information

5. Focused only on how to navigate, do not include other steps including extraction, analysis, opening website, closing environment, etc

Consider common web navigation patterns like:

- Searching for information

- Navigating through menus and links

- Going over the necessary pages

- Interacting with buttons and controls

**Example - Navigation Plan**

**Objective:**

Search for the product "Manhattan Comfort Avesta Double Side Table 2.0 Collection Free Standing Modern Side Table / TV Stand with Storage Includes 2 Doors with 3 Shelves and Features Splayed Legs, White/Stamp/Grey Legs" on the website. Navigate to the product page that contains detailed information including customer reviews. Ensure to collect review data that includes the rating scores or star ratings.

**Plan:**

1. Locate the search combobox labeled S̈earchïn the accessibility tree.

2. Input the product name: "Manhattan Comfort Avesta Double Side Table 2.0 Collection Free Standing Modern Side Table / TV Stand with Storage Includes 2 Doors with 3 Shelves and Features Splayed Legs, White/Stamp/Grey Legs´ into the search combobox.

3. Locate and activate the "Search" button to initiate the product search.

4. On the search results page, look for the product name or a closely matching link.

5. Click on the link corresponding to the desired product to navigate to its detailed product page.

6. On the product page, locate the section containing customer reviews or star ratings.

### A.1.3 INFORMATION EXTRACTION

During information extraction, we first instruct the LLM to select relevant observations based on the overall trajectory. The prompt is given as follows.

---

**Prompt - Relevant Observation Selection**

You are a judge agent in a web navigation and information seeking task.

Given a navigation objective (which includes the information to be found in the web environment) and a list of web navigation agent interaction history (with reason, action, and observation summary), select the step numbers that their observations are most likely to contain the information specified in the objective.

Analyze each step in one or two sentences. After this, return a JSON list of step numbers (e.g., [2, 5, 7]) that you believe contains the needed information in their observations. Note:

1) The action in a step will be executed and reflected in the observation in the next step. For example, if the action is 'click on the home page button', the observation in the next step will be the home page.

2) The action you see at each step may contain a number, like 'click[1316]'. This number is the index of the element in the observation. You may not know which element is clicked, but you can still use the reason to infer what that element is.

---

After selecting the relevant observations, we will first let the LLM to generate a prompt for extraction at each page. The reason for this step is to fix a data schema for easily integrating results from multiple pages.

---

**Prompt - Extraction Prompt Engineering**

You are an expert prompt engineer. Design a SINGLE prompt that, when shown together with a web-page text accessibility tree, makes another LLM extract and return ONLY a list of JSON object containing the fields that satisfy the user goal. Only extract the information specified in the user goal. Make sure each extracted entry also has one identifier field (add only one if there is no such key specified in user goal) that will helps accurate deduplication in the later stage. You need to specify 1) what information to be extracted, 2) what keys should be used for each JSON object in extracted list, 3) one simple example of the extracted JSON list. Make your prompt concise and only include these necessary infromation.

---

### A.1.4 EXECUTION

Below we provide the prompt for writing data analytic code during execution phase.

---

**Prompt - Data Analysis**

You are an analysis assistant that MUST write Python code.

---

You will be provided with objective and data samples (a small portion of all the data as a reference) for analysis as a reference.

• The data is pre-loaded in a variable named `data`.

• Assign your final answer to a variable named `answer`.

Return only one fenced block:

```python# code here

answer = ... ```

### A.1.5 RE-PLANNING

We provide the prompt of re-planning and one example here.

**Prompt - Re-planning**

You are a Dynamic Control Agent responsible for monitoring and adapting the task decomposition and navigation plan based on new observations during web navigation.

Your role is to:
1. Assess whether the current decomposition and navigation plan are still appropriate given the new web elements and information discovered
2. Determine if modifications are needed to better achieve the original objective
3. Update the decomposition and navigation plan when necessary

You will be provided with: - The original task objective - Current decomposition (Part 1 - Navigation, Part 2 - Analysis) - Current navigation plan - Current web page observation - Interaction history

Based on this information, you need to decide whether to:
- Keep the current decomposition and navigation plan unchanged
- Modify the decomposition to better reflect what needs to be done
- Update the navigation plan to account for new web elements or information discovered

Adhere to the following output format: {output_specifications}

Guidelines:
- Only modify decomposition/plan if you discover new web elements or information that significantly changes the approach
- Be conservative - don't change things unnecessarily
- Focus on practical improvements that will help achieve the objective more effectively
- Consider if new navigation paths or information sources have been discovered
- Ensure any updates are clear and actionable
- Do not include any task in analysis objective into the plan, as the plan is only for navigation

{website_tips}

**Example - Re-planning**

**Initial navigation plan**:

1. Navigate to the "Forum" tab on the website.

2. Click on "alphabetical" to view all forums listed alphabetically.

3. Visit each forum and locate submissions made by the user "thebelsnickle1991".

4. If there are multiple pages for submissions in a forum, continue navigating using the "More" link until reaching the last page.

**Re-planning reasoning:**

The current observation reveals that we are already on a page with submissions by the user "thebelsnickle1991." However, the original objective and decomposition specify starting from the "Forum" tab, then navigating to an alphabetical list of forums, and finally visiting pages containing submission data for this user. The current page already provides access to numerous submissions by the user, categorized under different forums, with a "More" link at the bottom for additional pages. This suggests a more direct path to fulfilling the objective.
The decomposition and navigation plan should be updated to prioritize directly extracting and analyzing the user's submissions from this page and subsequent pages accessed via the "More" link. This eliminates the need to navigate to the "Forum" tab or the alphabetical list of forums, streamlining the process.

**Updated navigation plan:**

1. Extract submission data for "thebelsnickle1991" directly from the user page.

2. Use the "More" link to navigate through additional pages containing submissions by "thebelsnickle1991" and extract data from those pages.

### A.1.6 OTHERS

Here we provide the prompt detail of the website tips we used and navigation specification for the navigation prompts above.

Following the WebChoreArena (Miyai et al., 2025), we used website tips for the evaluation in our experiments for our method and all the other baselines.

**Prompt - Website Tips**

Shopping

1. This website provides very detailed category of products. You can hover categories on the top menu to see subcategories.

2. If you need to find information about your previous purchases, you can go My Account > My Orders, and find order by date, order number, or any other available information

3. An order is considered out of delivery if it is marked as "processing" in the order status

4. When the task asks you to draft and email. DO NOT send the email. Just draft it and provide the content in the last message

5. If the review star rating is not directly available but the rating score is provided, you can estimate the star rating by dividing the rating score by 20. For example, a rating score of 80 corresponds to a 4-star review

6. Utilize the search if you need to find the information of a specific item, and use the top menu when you need to visit a category

**Shopping Admin**

Here are tips for using this website:

1. When you add a new product in the CATALOG > Products tab, you can click the downwardarrow beside the "Add Product" button to select options like "Simple Product", "Configurable Product", etc.

2. If you need to add new attribute values (e.g. size, color, etc) to a product, you can find the product at CATALOG > Products, search for the product, edit product with "Configurable Product" type, and use "Edit Configurations" to add the product with new attribute values. If the value that you want does not exist, you may need to add new values to the attribute.

3. If you need to add new values to product attributes (e.g. size, color, etc), you can visit STORES > Attributes > Product, find the attribute and click, and add value after clicking "Add Swatch" button.

4. You can generate various reports by using menus in the REPORTS tab. Select REPORTS > "report type", select options, and click "Show Report" to view report.

5. In this website, there is a UI that looks like a dropdown, but is just a 1-of-n selection menu. For example in REPORTS > Orders, if you select "Specified" Order Status, you will choose one from many options (e.g. Canceled, Closed, ...), but it's not dropdown, so your click will just highlight your selection (1-of-n select UI will not disappear).

6. Configurable products have some options that you can mark as "on" of "off". For example, the options may include "new", "sale", "eco collection", etc.

7. You can find all reviews and their counts in the store in MARKETING > User Content > All Reviews. If you see all reviews grouped by product, go REPORTS > By Products and search by Product name.

8. This website has been operating since 2022. So if you have to find a report for the entire history, you can select the date from Jan 1, 2022, to Today.

9. Do not export or download files, or try to open files. It will not work.

**Reddit**

Here are tips for using this website:

1. when the task mentions subreddit, it is referring to 'forum'

2. if you need find a relevant subreddit or forum, you can find the name after clicking "alphabetical" in the "Forum" tab

3. you can visit the next page with the link 'More', if the link 'More' is NOT visible in the current observation, this means you have reached the last page

### Gitlab

1. your user name is byteblaze

2. To add new members to the project, you can visit project information > members tab and click blue "invite members" button on top right

3. To set your status, click profile button on top right corner of the page (it's next to the question mark button) and click edit status

4. To edit your profile, click profile button on top right corner of the page (it's next to the question mark button) and click edit profile

5. You can also access to your information e.g. access token, notifications, ssh keys and more from "edit profile" page

6. Projects that you have contributed to are listed under Project / Yours / All tab of gitlab.site. You can sort repos using dropdown button on top right

7. Projects's repository tab has menus like Commits, Branches, Contributors, and more. Contributors tab shows contributors and their number of commits

8. If you want to see all the issues for you, you can either click button on the right of + icon on top right menu bar

9. When the task mentions branch main, it often means master

### Prompt - Navigation Specification

#### "click"

click [id]: To click on an element with its numerical ID on the webpage. E.g., 'click [7]' If clicking on a specific element doesn't trigger the transition to your desired web state, this is due to the element's lack of interactivity or GUI visibility. In such cases, move on to interact with OTHER similar or relevant elements INSTEAD.

#### "go_back"

go_back: To return to the previously viewed page.

#### "type"

type [id] [content] [press_enter_after=0/1]: To type content into a field with a specific ID. By default, the "Enter" key is pressed after typing unless 'press_enter_after' is set to 0. E.g., 'type [15] [Carnegie Mellon University] [1]' If you can't find what you're looking for on your first attempt, consider refining your search keywords by breaking them down or trying related terms.

> **"stop"**
>
> stop [answer]: To stop interaction and return response. ONLY use this action when you believe the objective is fully achieved and there is no need to furthur explore the website. Indicate the reason why you think the task objective has been completed within the brackets. E.g., 'stop [The review and rating information of all the products under electronic category has been tracked. There are 5 pages of products in total and all of them have been visited.] '

## A.2   IMPLEMENTATION DETAILS

### A.2.1   EXPERIEMENT DETAILS

In our main experiments, we utilize GPT-4o, GPT-5, GLM-4.5-air-fp8 as backbone models. For GPT-4o and GLM model, following AgentOccam, we utilize the same configuration, setting temperature as 0.5, top_p as 0.95. For GPT-5, we set reasoning effort to minimal, due to time and budget constraints.

We report results on four domains. Although the WebArena environment also contains a *Map* domain, we found that the service for this website was no longer accessible and therefore excluded it from evaluation. Moreover, since many multi-domain tasks involve the Map website, we also removed these tasks to ensure fair comparison with other methods that reported results only on the remaining domains.

We also did not compare with AgentSymbiotic (Zhang et al., 2025a) and Learn-by-Interact (Su et al., 2025), as the performance of these methods depends heavily on their proprietary retrieval-augmented generation (RAG) databases. Because neither of these works has released their databases, a direct comparison would not be fair or reproducible, and we therefore exclude them from our evaluation.

### A.2.2   NAVIGATION & EXECUTION

In our implementation, we follow the action selection mechanism introduced by AgentOccam (Yang et al., 2024a). Specifically, after the navigation agent generates candidate actions at each step (e.g., clicking an element, entering text, following a link, or stopping), we invoke a separate judge module to evaluate these candidates. The judge receives as input the task instruction, the current observation, the interaction history, and the candidate actions with their rationales. It then ranks or filters the candidates, selecting the action that is most consistent with the high-level objective.

This design allows the system to correct potential errors from the navigation agent. The judge therefore serves as a lightweight second-opinion layer, ensuring that the final action executed at each step is both safe and aligned with task goals.

During the final execution, if the task requires the analysis result as output, we directly output the analysis result. When writing the analysis code, if there is an error of executing the code, the agent will incorporate the error information and previous code to refine its response to generate another response. In the other case where the analysis results will be further used to complete web operations (*e.g.,* post a submission in Reddit), WEBDART will follow a similar mechanism as navigation, but with the analysis result in the context.

