# OpenReview forum: "WebDART: Dynamic Decomposition and Re-planning for Complex Web Tasks"
_ICLR.cc/2026/Conference — ICLR 2026 Conference Withdrawn Submission_

### Official Review · Reviewer_3GZA · 2025-10-27

**Soundness:** 3
**Presentation:** 3
**Contribution:** 3
**Rating:** 6
**Confidence:** 4

**Summary:**

A novel LLM agent framework (named, WebDART) is proposed to automate long-horizon complex web tasks, with hierarchical planning and adaptation. Sub-task decomposition allows the agents to dedicate their power to the allocated focus scope, which is designed with an in-depth analysis of the target problem and insights into human cognitive behaviors. At the same time, WebDART prompts the agent to dynamically re-plan based on their experiences by leveraging discovered shortcuts. The efficacy of WebDART is verified in two benchmarks: WebChoreArena and WebArena. The proposed method significantly outperforms the baselines across various LLM backbones in the first benchmark, and demonstrates remaining competency in the second testbed indicating a descent balance.

**Strengths:**

In this section, I demonstrate the strengths of this paper.

1. Solid motivation: The design behind the framework stems from grounded observations that complex tasks overload the common agent frameworks. The rationale behind the structural heuristics is also reasonably backed up by mentioning that the quality of sub-tasks differs from each other. The authors also point out the possible limitations, revealing the necessity of the second component (i.e., adaptation).
2. Empirical supports: While navigating all the possible pages to get the task-relevant information is demanding, it can often be exhaustive. Table 2 demonstrates that such limitations can be significantly overcome with their proposed method. These results strongly support the effectiveness of the design of WebDART.
3. Case study analysis: The case study analysis allows readers to understand how WebDART performs well in practice. The readability of the study is high, as it is organized as a concise table.

**Weaknesses:**

Here, I present the weaknesses/questions/suggestions of this work.

1. Missing discussions/references: In the related work section, as this work focuses on prompting-based agent frameworks, discussions on possible strengths/limitations compared to fine-tuning methods would make the paper more solid. Currently, the authors compare with AgentSymbiotic, as a representative of finetuning-based methods, but with a lack of depth. I provide several finetuning-based methods, which I hope will be discussed [1,2,3]. To clarify, comparisons with these agents in experiments don’t seem very demanding.
2. Cost: How much did it cost to run all the experiments, including the baselines? I believe that cost information allows comparing the compute resources used between the baselines, as well as easy estimation of requirements when experimenting with WebDART as a baseline for other research.
3. Analysis on multi-agent baselines: I think more comparisons with multi-agent frameworks can be included. Mainly, it’d be interesting to see where the main differentiation towards success arises in the pipeline, compared to other multi-agent frameworks.
4. Marginal improvements on WebArena: While the authors stated that many tasks do not demand complex sub-task planning in WebArena, it is still questionable why the WebDART agents do not gain much in this benchmark, as this phenomenon signals a possibility of biased design in the WebChoreArena benchmark. To be fair, there are stronger baselines in WebArena [4]. I think that the authors should discuss more to clarify this, as (at least) discussing what improvements in WebDART can make it outperform the baselines. Also, the “bypassing” mechanism should be elaborated.


References:

[1] Qi et al., “WebRL: Training LLM Web Agents via Self-Evolving Online Curriculum Reinforcement Learning” (ICLR 2025).

[2] Lee et al., “Learning to contextualize web pages for enhanced decision making by LLM agents” (ICLR 2025).

[3] Qin et al., “UI-TARS: Pioneering Automated GUI Interaction with Native Agents” (preprint 2025).

[4] https://webarena.dev/.

**Questions:**

Questions and suggestions for the authors are listed in the above weaknesses section for brevity.

---

> ### Author Response · Authors · 2025-11-25
> **Response to Reviewer 3GZA (1/2)**
>
> We thank Reviewer 3GZA for the insightful comments. We provide our comments as below:
>
> **[Q1] Missing discussions/references.**
>
> Thank you for pointing out these important related works. We had updated our manuscript to include them in the related work discussion. We also include the discussion here for your convenience.
>
> Finetuned web agents represent another important line of work complementary to training-free designs like ours. These approaches explicitly fine-tune an LLM policy using domain-specific trajectories to encode stronger priors for multi-step decision making. Recent examples include curriculum-based reinforcement learning agents that evolve their own training distribution over time [1], models that learn webpage-specific contextualization layers to filter DOM observations before acting [2], and GUI-generalist agents, finetuned on large multimodal UI demonstrations, to perform precise manipulation and element grounding [3]. While fine-tuning often yields higher in-distribution accuracy, these methods typically require expensive data generation and can be brittle under distribution shifts. In contrast, our approach instead relies on structured task decomposition and interface optimization to achieve strong generalization without additional training cost.
>
> **[Q2] Detailed Compute and Token Cost Analysis for WebDART and Baselines.**
>
> Thank you for the question. We agree that reporting experiment cost is important for understanding the efficiency improvement. Below, we provide a detailed **breakdown of the token consumption and cost** **on WebChoreArena** for WebDART and AgentOccam, which is the most comparable baseline in terms of action space, observation format, and overall workflow.
>
> To ensure consistency, we report costs using **GPT-4o**, because GPT-5 includes implicit reasoning traces that make token usage difficult to estimate reliably. As of the time of writing, GPT-4o is priced at **2.5 USD per million input (prefill) tokens** and **10 USD per million output tokens**.
>
> The total number of steps, input tokens, output tokens, and resulting compute costs are shown below:
>
> **[Table Q2]. Token usage and compute cost for WebDART and AgentOccam using GPT-4o on WebChoreArena.**
>
> |            | Steps | Input tokens | Output tokens | Cost      |
> | ---------- | ----- | ------------ | ------------- | --------- |
> | AgentOccam | 22.1  | 86.4 M       | 6.1 M         | 277.0 USD |
> | WebDART    | 19.6  | 83.8 M       | 6.0 M         | 269.5 USD |
>
> The results show that, despite including more modules (navigation, information extraction, and analysis), WebDART is **more cost-efficient**, requiring fewer steps and marginally fewer tokens overall while still achieving significantly higher accuracy.
>
> **[Q3] Analysis on multi-agent baselines.**
>
> Thank you for the suggestion. We agree that comparing WebDART with existing multi-agent frameworks provides useful insight.
>
> We include **WebPilot**, a representative multi-agent method for comparison. It separates high-level reasoning from low-level execution by using: (1) **a planner model** to generate step-by-step intentions, and (2) **an executor model** to convert these intentions into concrete browser actions. WebPilot also employs a **self-correction module** to detect inconsistencies between the expected and actual webpage state and attempt recovery.
>
> Since WebPilot does not provide an open-source implementation, we reproduced it for evaluation on WebChoreArena. Please note that there are many hyper-parameters (e.g., reward weights, scroll budget, and incomplete MCTS parameters) in WebPilot that were not disclosed by the original paper, and we set these hyper-parameters using a reward weight of [0.5, 0.5], scroll budget of 5, and and selected reasonable defaults for the missing MCTS components, such as normalizing rewards to [0,1]. The experimental results with GPT-4o as backbone are presented below.
>
> ##### **[Table Q3]. Comparison between WebPilot (multi-agent) and WebDART (GPT-4o backbone)**
>
> |               | Shopping | Admin | Reddit | Gitlab | Avg  |
> | ------------- | -------- | ----- | ------ | ------ | ---- |
> | WebChoreArena |          |       |        |        |      |
> | SteP          | 2.6      | 0.0   | 0.0    | 4.7    | 1.8  |
> | BrowserGym    | 0.9      | 5.5   | 2.3    | 3.9    | 3.2  |
> | AWM           | 3.4      | 8.8   | 4.5    | 4.7    | 5.4  |
> | AgentOccam    | 10.3     | 9.9   | 4.5    | 7.1    | 8.0  |
> | **WebPilot**  | 6.8      | 2.2   | 4.5    | 3.9    | 4.4  |
> | WebDART       | 18.8     | 19.8  | 12.9   | 9.4    | 15.2 |
>
> We highlight that **WebDART outperforms the multi-agent baseline and achieves the highest performance in every domain**. This further verifies the effectiveness of our method.

---

> ### Author Response · Authors · 2025-11-25
> **Response to Reviewer 3GZA (2/2)**
>
> **[Q4] Why is the improvement on WebArena marginal? Why are some stronger baselines not included for comparison?**
>
> Thank you for the question. We address the two components separately for clarity.
>
> 1. Why are WebDART’s improvements on WebArena relatively small?
>
> The purpose of the WebArena experiments is **not** to highlight large performance gains, but rather to demonstrate that **WebDART generalizes well to simple navigation tasks** without sacrificing performance.
>
> WebArena primarily contains **short, low-complexity navigation tasks**, where most competent agents already perform reasonably well. Because WebDART is designed to address **complex, multi-step, and constraint-heavy tasks** in WebChoreArena, its advantages are naturally less significant there. Despite this, our method still achieves competitive performance, demonstrating its generalizability. At the meantime, it brings more performance gain on more complex tasks.
>
> 2. Why were some top-performing baselines not included?
>
> We based our comparisons on **open-sourced** baselines that can be evaluated **fairly and reproducibly**. At the time of our submission, the top-performing WebArena agents were: AgentSymbiotic, Learn-by-Interact, AgentOccam. However:
>
> - **AgentSymbiotic** and **Learn-by-Interact** are **retrieval-augmented** methods with a pre-collected exploration trajectories on similar tasks. At each step of a task, they retrieve similar trajectories as demonstrations, which introduces an unfair advantage and potential data contamination. Also, their retrieval data is not open-sourced, making fair and reproducible comparison infeasible (as discussed in Appendix A.2.1 of our submission). Therefore, we exclude the two methods.
> - Another method**, IBM CUGA**, which achieves stronger performance, released its code *after* our submission deadline. Moreover, it relies on **web MCP tool use**, which introduces capabilities not available to other baselines or to WebDART, making direct comparison difficult without mismatched assumptions.
> - **AgentOccam** is the strongest fully open-sourced agent available at submission time.
>    Starting from AgentOccam, we incorporated all powerful baselines that could be fairly evaluated.
>
> Importantly, WebDART is **model-agnostic**: it can integrate seamlessly with any strong navigation agent on the WebArena leaderboard. This is because WebDART focuses on **task decomposition**, not on replacing the navigation module. Thus, as more SOTA agents become available, they can be plugged into WebDART to further improve performance.
>
> We appreciate the reviewer’s suggestion and will include a more detailed discussion of the WebArena leaderboard and experimental choices in the final version.
>
> Reference:
>
> [1] Qi, Zehan, et al. "Webrl: Training llm web agents via self-evolving online curriculum reinforcement learning." arXiv preprint arXiv:2411.02337 (2024).
>
> [2] Lee, Dongjun, et al. "Learning to contextualize web pages for enhanced decision making by LLM agents." arXiv preprint arXiv:2503.10689 (2025).
>
> [3] Qin, Yujia, et al. "Ui-tars: Pioneering automated gui interaction with native agents." arXiv preprint arXiv:2501.12326 (2025).

---

> ### Author Response · Authors · 2025-11-26
>
> Dear Reviewer 3GZA,
>
> We thank you for the time on reviewing and the constructive feedback again. We really hope to discuss further with you to see if our response answers your questions.
>
> We genuinely hope reviewer 3GZA could kindly check our response. Thank you very much!

---

> > ### Comment · Reviewer_3GZA · 2025-11-28
> >
> > I remain positive on this work. At the same time, I do believe that this work can be significantly improved. I leave some comments and remaining concerns to be resolved (with actionable suggestions). I maintain my rating at 6.
> >
> > 1. Missing discussions/references: I am pleased to see that this is helpful for your work.
> > 2. Detailed Compute and Token Cost Analysis: I am grateful to see a detailed analysis of the compute. I believe this is indeed a positive signal, and hope the future version includes more analysis on other baselines.
> > 3. Multi-agent baselines: I agree that the new experimental results further verify the effectiveness of the proposed method.
> > 4. On WebArena: I still can’t understand that WebDART has marginal advantages over shorter, simpler tasks. Could the authors explain why WebDART fails on the remaining 50% of the tasks in WebArena? Continuing from Q3, can the authors explain where WebDART shows success differently from the other baselines?  Furthermore, I suggest that to show the performances in other benchmarks that have different characteristics (such as WorkArena [5]), as pointed by the Reviewer bz3S as W3. I believe that this can provide strong evidence that the WebDART is not overfit to WebChoreArena.
> >
> > [5] A. Dourin, “WorkArena: How Capable Are Web Agents at Solving Common Knowledge Work Tasks?” (ICML 2024).

---

### Official Review · Reviewer_RREB · 2025-10-29

**Soundness:** 3
**Presentation:** 3
**Contribution:** 2
**Rating:** 4
**Confidence:** 4

**Summary:**

This paper introduces WEBDART, a framework that enables large language model (LLM) agents to handle complex web tasks that require long-horizon reasoning and structured exploration. It dynamically decomposes each task into three subtasks: (1) navigation, (2) information extraction, and (3) execution, allowing the model to focus on one ability at a time. During navigation, the agent adaptively re-plans its strategy when new filters or interface shortcuts appear, reducing redundant actions and improving efficiency. This modular and adaptive design enhances task completion and robustness in complex web environments while maintaining strong performance on simpler tasks. Overall, WEBDART demonstrates that dynamic decomposition and real-time re-planning can significantly improve the reasoning and adaptability of LLM-based web agents.

**Strengths:**

- **Well-Justified Motivation**: The paper effectively addresses the importance of long-horizon web tasks as a fundamental challenge in current web-agent research.

- **Clear Writing and Organization**: The paper is well-written and easy to follow, with a well-organized structure and clear presentation of the proposed approach.

- **Simple Yet Effective Design**:This paper employs an intuitive three-stage decomposition that mirrors how humans naturally approach complex web tasks, resulting in a method that is both easy to understand and practically effective.

**Weaknesses:**

- **Lack of empirical justification for the conservative decomposition scheme**: The paper adopts the conservative scheme (deferring constraint handling to later stages) as the default strategy. However, this design choice is not supported by any preliminary analysis, empirical comparison, or prior evidence—for example, there is no ablation or user study contrasting conservative versus tightly coupled decompositions. Given that the efficiency of each scheme “hinges on site features” (line 204), a fixed conservative default appears heuristic rather than data-driven, and its general validity across domains remains unclear. A short pilot experiment or reference to earlier literature on adaptive task partitioning would strengthen this methodological decision and clarify why the conservative bias is justified beyond intuition.

- **Heuristic Nature of Information Extraction**: The information extraction pipeline is heuristic, relying on LLM prompts to select relevant pages and extract fields without any quantitative validation or ablation. The paper explains that the model “returns an index set that marks the pages most likely to contain the required information,” yet provides no concrete mechanism or evidence to show how reliable this selection is. Furthermore, the dismissal of the LLM-generated parser baseline is entirely qualitative, lacking any comparative results or failure statistics. Overall, the decision to rely solely on prompt-based extraction appears intuitive rather than experimentally justified, leaving uncertainty about its robustness and reproducibility across diverse web structures.

- **Lack of In-depth Performance Analysis**: While the paper reports overall success rates on the WebChoreArena benchmark [1], it does not provide finer-grained analyses that could strengthen its empirical claims. In the original benchmark, performance is typically broken down by cross-site domains as well as by task types such as Calculate, Long-Term Memory, Massive Memory, and Other. However, WEBDART’s results are aggregated, making it unclear which categories drive the observed improvements. The absence of such detailed breakdowns limits the interpretability of the reported gains and prevents deeper insights into where the proposed method truly excels or struggles.

[1] Miyai, Atsuyuki, et al. "WebChoreArena: Evaluating Web Browsing Agents on Realistic Tedious Web Tasks." arXiv preprint arXiv:2506.01952 (2025).

**Questions:**

- In Section 4.2, the authors claim that the observed improvements “highlight the advantage of shifting constraint handling to the data analysis stage.” However, it is unclear how the empirical results in Table 1 specifically support this interpretation. Could the authors clarify what evidence connects the performance gains to this design choice?

- Table 3 reports the Results on the WebArena benchmark and includes additional baselines such as HybridAgent [1] and WebPilot [2], which show competitive performance. How do these baselines perform on WebChoreArena, and were they excluded due to reproducibility constraints or unavailability of results?

- As an ablation, how does performance change when the routing module is disabled, particularly on the WebArena benchmark? It would be helpful to know how much accuracy drops and what types of routing errors occur (e.g., skipping extraction when it is actually required). Additionally, could the authors provide a brief analysis of the common failure cases in WEBDART?

[1] Song, et al. "Beyond browsing: Api-based web agents." arXiv preprint arXiv:2410.16464 (2024).

[2] Zhang, et al. "Webpilot: A versatile and autonomous multi-agent system for web task execution with strategic exploration." AAAI 2025

---

> ### Author Response · Authors · 2025-11-25
> **Response to Reviewer RREB (1/4)**
>
> We thank Reviewer RREB for the valuable feedback. Please see our response to the questions and major comments below.
>
> **[Q1] Lack of empirical justification for the conservative decomposition scheme.**
>
> Thank you for your question. We would like to justify our conservative decomposition scheme from two aspects: (1) **motivation from LLM capacity mismatch in complex web environments** and (2) **empirical comparison with a tightly coupled decomposition scheme**.
>
> - The motivation for starting with a conservative decomposition comes directly from our observations of LLM-based web agents.
>
>   - First, we find a clear capability disparity: **LLMs are substantially stronger at information extraction and data analytics** (e.g., coding) than at web navigation. This pattern is consistent with prior evaluations in Multi-hop QA, function-level code generation, and web navigation. While modern LLMs have nearly saturated performance on the first two categories, there remains significant room for improvement in web navigation. For instance, GPT-4o obtains 90.2% accuracy on HumanEval but only 11.5% accuracy on WebArena-style navigation tasks when used without an agent workflow.
>   - Second, the structure of real web environments further amplifies this mismatch. Web pages often contain numerous interactive elements that encode nontrivial data constraints (e.g., nested filters). When agents attempt to resolve complex constraints directly within navigation, they become entangled in these interactions and consequently reveal the backbone LLM’s weakness in fine-grained navigation.
>
>   These observations motivate the conservative decomposition design: **reduce the complexity of navigation subtasks as much as possible and shift more responsibility to information extraction and data analytics**, where LLMs are demonstrably more capable.
>
> - To further validate this decision, we also implemented our method using a constraint-aware decomposition contrary to the conservative decomposition, where **constraints were explicitly included into the navigation objective**. For example, consider the navigation goal of *“Please tell me the full name of the product with the most reviews among the highest-rated products in the highest 20% price range of Rugs, Pads & Protectors.”*. Under the constraint-aware decomposition, the navigation objective will be *“Visit product listing under Rugs, Pads & Protectors for products, sorted by rating, number of reviews, and price within range of highest 20%”*. In other words, the navigation goal still explicitly includes all constraints from the original task. We compare the performance of our method with two decomposition schemes as below:
>
> **[Table Q1]. Performance comparison of different decomposition strategies. (GPT-4o backbone)**
>
> |                                | Shopping | Reddit | Admin | Gitlab | Avg  |
> | ------------------------------ | -------- | ------ | ----- | ------ | ---- |
> | Conservative Decomposition     | 18.8     | 19.8   | 12.9  | 9.4    | 15.2 |
> | Constraint-aware Decomposition | 12.8     | 15.4   | 10.6  | 8.7    | 11.9 |
>
>
>
> - As the table shows, the conservative decomposition consistently outperforms the constraint-aware variant across all domains. This is because the constraint-aware approach includes constraints such as filters and sorting directly into the navigation objective, forcing the agent to manipulate price-range controls, rating sorters, and other UI elements. These interactions frequently cause navigation failures. In contrast, the conservative scheme avoids these brittle operations by deferring constraints to the analysis stage, where LLMs are far more reliable.
>
>   Overall, this comparison demonstrates that the conservative strategy is not just a heuristic preference but a more robust choice, yielding a 3.3-point average improvement across domains.

---

> ### Author Response · Authors · 2025-11-25
> **Response to Reviewer RREB (2/4)**
>
> **[Q2] Heuristic Nature of Information Extraction.**
>
> Thank you for raising this concern. **To clarify the reliability of our information extraction pipeline, we provide quantitative evaluations for both (i) the key-page selection step and (ii) the extraction method itself.**
>
> - **Reliability of key-page selection.** To evaluate whether the key-page selection step is reliable, we measured its accuracy in the shopping domain using human-annotated ground truth. A correct navigation is defined as an agent visiting all the pages recording the needed information, and a correct selection is defined as the page selected by the agents including all the pages containing necessary information. The results are summarized below:
>
> **[Table Q2.1]. Analysis of web page selection.**
>
> | Total Task # | Correct Navigation # | Correct Selection # | Selection Acc. |
> | ------------ | -------------------- | ------------------- | -------------- |
> | 117          | 67                   | 64                  | 95.5%          |
>
> These results show that once the agent navigates to the correct region of the website, the LLM almost always identifies the correct informative pages, achieving a selection accuracy of more than 95%. This demonstrates that the mechanism operates consistently and is empirically reliable rather than an unvalidated heuristic.
>
> - **Reliability of prompting-based information extraction.** Our decision to use prompting-based extraction over an LLM-generated coding (parser) approach stems from two empirical observations::
>
>   - **First, prompt-based extraction is more stable.** We experimented with both coding-based and prompting-based extraction methods. In practice, the coding-based approach was highly unstable due to the heterogeneous and complex structure of accessibility trees across real web pages. LLMs frequently failed to generate correct parsing code, leading to brittle extraction pipelines.
>
>     To quantify this, we sampled 1,000 webpages from the WebArena environment and defined an extraction objective for each page to construct a dedicated evaluation set. We tested both extraction strategies with identical objectives. An extraction was counted as valid if the returned JSON object was non-empty. The validity results are reported below:
>
>     **[Table Q2.2]. The validity comparison of different information extraction strategies.**
>
>     |          | Coding extraction | Prompting extraction |
>     | -------- | --------------------- | ------------------------ |
>     | Validity | 94.4%                 | 99.0%                    |
>
>     The prompting-based method achieves substantially higher reliability, demonstrating that its robustness is not merely intuitive but quantitatively supported.
>
>   - **Second, prompt-based extraction yields better generalizability.** Our current agent observes purely text-based accessibility trees. However, future versions of the agent may incorporate webpage screenshots or multimodal inputs. In such settings, a coding-based extraction approach would no longer be feasible, whereas prompting-based extraction naturally generalizes to multimodal inputs. This provides an additional practical motivation for choosing the prompting-based approach.

---

> ### Author Response · Authors · 2025-11-25
> **Response to Reviewer RREB (3/4)**
>
> **[Q3] Lack of in-depth performance analysis across task categories.**
>
> Thank you for this valuable suggestion. We agree that breaking down performance by task type provides deeper insight into where our method excels and where challenges remain. In response, we computed category-level results of GPT-5 following the original WebChoreArena taxonomy: **Calculate**, **Long-Term Memory**, **Massive Memory**, and **Other.** We also include the AgentOccam baseline to more clearly illustrate the performance gains contributed by Our method.
>
> **[Table Q3]. Performance breakdown across task categories. Percentages in parentheses denote the proportion of tasks in each category.**
>
> **(Note: the “Other” category constitutes only 9% of the benchmark, so differences here have limited impact on overall performance.)**
>
> |                | Massive Memory (28.1%) | Long-term Memory (41.5%) | Calculation (21.4%) | Other (9.0%) |
> | -------------- | ---------------------- | ------------------------ | ------------------- | ------------ |
> | AgentOccam     | 26.0                   | 23.7                     | 26.0                | 42.9         |
> | WebDART (ours) | 33.6                   | 28.4                     | 35.0                | 31.0         |
>
> We highlight several observations.
>  **First**, across the three major categories, including *Massive Memory*, *Long-Term Memory*, and *Calculation*, which collectively account for over 90% of the benchmark, WebDART consistently achieves substantial improvements over the baseline method, AgentOccam. This demonstrates that the proposed decomposition framework yields broad and meaningful gains.
>
> **Second**, we find that *Long-Term Memory* tasks remain the most challenging. These tasks emphasize long-horizon **navigation**, an ability where current LLMs are comparatively weaker. In contrast, *Massive Memory* tasks rely more heavily on **information extraction and processing**, and *Calculation* tasks emphasize **data analysis**. These areas are where LLMs perform more strongly. Accordingly, WebDART yields its largest improvements in these two categories.
>
> **Finally**, although AgentOccam slightly outperforms our method on the *Other* category, this category represents only 9% of total tasks and contains a diverse mixture of problem types with limited statistical weight. As such, its small performance fluctuation has minimal effect on the overall conclusions.
>
> **[Q4] How do WebPilot and HybridAgent perform on WebChoreArena? Are they excluded due to reproducibility constraints or unavailability of results?**
>
> Thank you for your question. We clarify the details for WebPilot and HybridAgent as below.
>
> - **First**, we did not include these two methods on WebChoreArena because: (1) **WebPilot is not open-sourced for comparison**, and (2) **HybridAgent adopts a totally different navigation strategy**. Specifically, HybridAgent assumes that the website provides a set of APIs to complete tasks, rather than interacting with on-page elements (e.g., clicking buttons, opening menus, following links) for navigation. This API-based interaction gives HybridAgent capabilities that are not available to other baselines and our method, leading to unfair advantage. Moreover, WebChoreArena does not expose such APIs, **making HybridAgent not applicable in this setting**. For this reason, we exclude it on WebChoreArena.
>
>   Please note that the WebArena results for both baselines are available on the official leaderboard, so we report those directly.
>
> - **Second**, although WebPilot does not provide an open-source implementation, **we followed your suggestion and reproduced it** for evaluation on WebChoreArena. Please note that there are many hyper-parameters (e.g., reward weights, scroll budget, and incomplete MCTS parameters) in WebPilot that were not disclosed by the original paper, and we set these hyper-parameters using a reward weight of [0.5, 0.5], scroll budget of 5, and and selected reasonable defaults for the missing MCTS components, such as normalizing rewards to [0,1]. The experimental results with GPT-4o as backbone are presented below.
>
>   **[Table Q4]. Performance of WebPilot and baselines on WebChoreArena (GPT-4o backbone).**
>
>   |            | Shopping | Reddit | Admin | Gitlab | Overall |
>   | ---------- | -------- | ------ | ----- | ------ | ------- |
>   | Browsergym | 0.9      | 5.5    | 2.3   | 3.9    | 3.2     |
>   | WebPilot   | 6.8      | 2.2    | 4.5   | 3.9    | 4.4     |
>   | AgentOccam | 10.3     | 9.9    | 4.5   | 7.1    | 8.0     |
>   | WebDART    | 18.8     | 19.8   | 12.9  | 9.4    | 15.2    |
>
>   We highlight that **WebDART achieves the highest performance in every domain**, with especially large gains on Shopping and Reddit. This further verifies the effectiveness of our method.

---

> ### Author Response · Authors · 2025-11-25
> **Response to Reviewer RREB (4/4)**
>
> **[Q5] The effect of routing.**
>
> Since tasks in WebArena are mainly simple tasks which can be solved by using only the navigation module, activating extra modules like information extraction may introduce a failure. We disable the routing here to only let the agent conduct one turn of navigation.
>
> ##### **[Table Q5]. Performance of WebDART on WebArena with and without routing (GPT-4o backbone).**
>
> |                       | Shopping | Reddit | Admin | Gitlab | Overall |
> | --------------------- | -------- | ------ | ----- | ------ | ------- |
> | WebDART               | 36.0     | 67.9   | 41.2  | 47.2   | 48.1    |
> | WebDART (w/o routing) | 36.0     | 69.9   | 44.3  | 47.2   | 49.4    |
>
> From the results above, we found naively disabling the routing can even improve the performance of WebDART on WebArena, because the decomposition error is mitigated for a small amount of the tasks. The slight performance difference here can be attributed to the tradeoff of generalizing our WebDART on both complex web tasks and simple web navigation tasks.
>
> **[Q6] Analysis of the common failure cases.**
>
> Thank you for the question. To understand where errors originate within WebDART, we conducted a quantitative analysis over **100 randomly sampled trajectories**. For each trajectory, we identified only the **earliest point of failure**, since downstream errors are typically propagated from earlier mistakes. The distribution of failure sources is shown below.
>
> ##### **[Table Q6]. Failure analysis of different modules.**
>
> |         | Decomposition | Navigation | Information Extraction | Execution |
> | ------- | ------------- | ---------- | ---------------------- | --------- |
> | Failure | 7.8%          | 59.3%      | 21.9%                  | 10.9%     |
>
> These results indicate that **navigation remains the dominant bottleneck**, accounting for nearly 60% of failures. This aligns with our core motivation: current LLMs are substantially less capable at fine-grained, multi-step web navigation compared to information processing and analytics.
>
> Regarding navigation-related failure modes, the most frequent navigation failures include:
>
> - **Forgetting earlier steps**. The agent sometimes forgets previously gathered context or loses track of the website’s structure, leading it to take incorrect paths or fail to locate required information.
> - **Stopping too early.** The agent stops navigating before collecting sufficient information, causing incomplete or incorrect downstream reasoning.
> - **Navigation to irrelevant or misleading pages**. The agent occasionally follows distractor links or navigates into side branches, which then misguides the information extraction module.
>
> These observations confirm that navigation remains the primary challenge for LLM-based web agents and reinforce the motivation behind WebDART’s conservative decomposition strategy.

---

> ### Author Response · Authors · 2025-11-26
>
> Dear Reviewer RREB,
>
> We thank you for the time on reviewing and the constructive feedback again. We really hope to discuss further with you to see if our response answers your questions.
>
> We genuinely hope reviewer RREB could kindly check our response. Thank you very much!

---

### Official Review · Reviewer_bz3S · 2025-11-01

**Soundness:** 3
**Presentation:** 3
**Contribution:** 2
**Rating:** 2
**Confidence:** 4

**Summary:**

This paper introduces a training-free framework that improves LLM-based web agents by dividing complex objectives into navigation, information extraction, and execution subtasks while dynamically revising plans as new web elements appear. This modular and adaptive design allows the agent to focus on one skill at a time and adjust strategies in real time, leading to up to 13.7% higher accuracy and 14.7 fewer navigation steps on complex benchmarks, without sacrificing performance on simpler tasks.

**Strengths:**

S1) Clarity and Organization
- The paper is clearly written and well structured. It effectively identifies the limitations of existing web tasks and presents a complex yet well-justified task definition along with a corresponding solution. The modular framework design and clear categorization of sub-tasks make the methodology easy to interpret. Figures and tables are informative and greatly aid understanding of the workflow.

S2) Technical Soundness and Contribution
- The overall structure and writing are coherent, and the paper presents a convincing motivation for introducing adaptive re-planning, emphasizing its necessity within dynamic web navigation. The proposed WebDART methodology is applied successfully, demonstrating meaningful improvements over existing baselines.

S3) Competitive Performance
- The approach achieves strong quantitative results, showing robustness across different web-based benchmarks and supporting the validity of the proposed framework.

**Weaknesses:**

W1) Lack of Novelty
- The proposed approach primarily integrates existing techniques rather than introducing a fundamentally new concept. While the composition of prior methods is well executed, the paper lacks clear methodological or theoretical innovation that distinguishes it from prior work.
- Moreover, the claimed design motivation—being inspired by human web search behavior—lacks supporting evidence from prior studies or pilot experiments. Including such references or empirical validation would strengthen this claim.

W2) High Computational and Monetary Cost
- The framework’s multi-stage navigation process, including dynamic re-planning decisions, likely leads to frequent LLM calls and thus high computational and monetary costs. To substantiate the framework’s practical usability, the paper should include quantitative evidence such as the number of LLM invocations per episode or the overall inference cost, along with a discussion of trade-offs between performance and efficiency.
- An efficiency analysis comparing total inference cost, time, or action steps with other methods would be valuable, especially since different baselines may not define navigation steps equivalently.

W3) Limited Generalization and Evaluation Scope
- The framework heuristically decomposes web tasks into three sub-tasks, but it remains unclear whether this decomposition generalizes across diverse web task categories.
- Experiments are conducted only on two benchmarks—WebArena and WebChoreArena—which, despite differing in complexity, share similar task structures. Consequently, the evaluation does not sufficiently demonstrate robustness to broader and more heterogeneous web task types.
- The authors should evaluate the framework on at least one additional web-based agent task (e.g., GAIA [2] or SimpleQA [3]), as the current setting appears tailored to the WebArena family.
- In addition, the prompt design for navigation planning (“navigating through menus and links,” “interacting with buttons and controls”) imposes a strong prior, raising further concerns about generalization capability.

W4) Outdated Baseline Comparison
- In Tables 1 and 2, most baseline performances are cited from prior studies, resulting in comparisons primarily against older methods.
- To more convincingly demonstrate the proposed approach’s effectiveness, the paper should include comparisons with more recent web-agent methodologies (for example, WebWalker [1]).

W5) Lack of Analytical Depth
- The current analysis is shallow. Reporting only accuracy and average steps provides limited insight.
- Additional ablation studies—such as the number of re-planning events, subtasks per instruction, their distribution, and the effect of fast-path routing—would offer deeper analytical value and better explain the framework’s behavior under different task conditions.
References

[1] Wu et al. WebWalker: Benchmarking LLMs in Web Traversal. Arxiv preprint. 2025
[2] Mialon et al. GAIA: A Benchmark for General AI Assistants. ICLR 2023
[3] Jason et al. Measuring short-form factuality in large language models. Arxiv preprint. 2024

**Questions:**

See the weaknesses

---

> ### Author Response · Authors · 2025-11-25
> **Response to Reviewer bz3S (1/3)**
>
> We thank Reviewer bz3S for the feedback. Please see our response below.
>
> **[Q1]** **The proposed approach primarily integrates existing techniques rather than introducing a fundamentally new concept?**
>
> We politely disagree with this comment, and we would like to clarify the novelty of our method as below.
>
> - Our main contribution is not proposing modules such as navigation or information extraction, but decoupling the sub-tasks that require different capabilities, instead of traditional GUI-based web agents that couple different capabilities solely during navigation. Although some GUI-based web agents may include a decomposition module (e.g., WebPilot includes a planner agent that decomposes the original task into several sub-tasks), such decomposition is fundamentally different from ours, as it mainly focuses on breaking down a long-horizon task into several short-horizon tasks that require the same capability. For example, a task such as “Create a ‘NolanFans’ repo, listing Nolan’s Oscar-winning films in a README file” can be divided into three sub-tasks — “Go to posts discussing Nolan’s Oscar-winning films,” “Create a ‘NolanFans’ repo in GitLab,” and “Create a README file listing Nolan’s Oscar-winning films.” All of these sub-tasks are navigation tasks, just like the original task. In contrast, our method focuses on decomposing the task at the required capability level, rather than at the task-horizon level.
> - Another main contribution is our solution for how to more reasonably decompose a task. We observed that when decomposing a complex task into sub-tasks with different capabilities, the granularity/responsibility of each sub-task can be difficult to determine because: 1) decomposition at the beginning lacks prior knowledge about the environment (e.g., whether there is a usable shortcut to make navigation easier); and 2) backbone LLMs’ capabilities vary — they are usually strong at data analytics but weak at web navigation. Based on this observation, we proposed our dynamic decomposition mechanism, which leads to improvements in both efficiency and performance on the WebArena and WebChoreArena benchmarks.
>
> - We revisited the SOTA GUI-based web agents currently in WebArena and found that almost all of them still focus solely on the navigation module and do not decompose the task based on different agent capabilities as we do. For example, Claude Code + GBOX MCP converts natural-language goals into step-by-step web actions by interpreting user intent and executing the required UI operations on the target website.
>
>   This further supports our claim regarding our contribution, and our work is not merely an integration of existing methods. If you still believe so, we respectfully request that you provide “the existing work” as a reference, as other reviewers did (e.g., Reviewer 3GZA). **Please note that the related work of GUI-based web agents and search-based web agents should not be mixed together here (please see our more detailed explanation in Q3).** In addition, it would be helpful for us to better interpret this review if the reviewer could provide his/her definition of “a fundamentally new concept” and the related web agent work that fits this definition.
>
> **[Q2]** **High Computational and Monetary Cost.**
>
> We agree that reporting experiment cost is important for understanding the efficiency improvement. Below, we provide a detailed **breakdown of the token consumption and cost** **on WebChoreArena** for WebDART and AgentOccam, which is the most comparable baseline in terms of action space, observation format, and overall workflow.
>
> To ensure consistency, we report costs using **GPT-4o**, because GPT-5 includes implicit reasoning traces that make token usage difficult to estimate reliably. As of the time of writing, GPT-4o is priced at **2.5 USD per million input (prefill) tokens** and **10 USD per million output tokens**.
>
> The total number of steps, input tokens, output tokens, and resulting compute costs are shown below:
>
> **[Table Q2]. Token usage and compute cost for WebDART and AgentOccam using GPT-4o on WebChoreArena.**
>
> |            | Steps | Input tokens | Output tokens | Cost      |
> | ---------- | ----- | ------------ | ------------- | --------- |
> | AgentOccam | 22.1  | 86.4 M       | 6.1 M         | 277.0 USD |
> | WebDART    | 19.6  | 83.8 M       | 6.0 M         | 269.5 USD |
>
> The results show that, despite including more modules (navigation, information extraction, and analysis), WebDART is **more cost-efficient**, requiring fewer steps and marginally fewer tokens overall while still achieving significantly higher accuracy.

---

> ### Author Response · Authors · 2025-11-25
> **Response to Reviewer bz3S (2/3)**
>
> **[Q3] The authors should evaluate the framework on at least one additional web-based agent task (e.g., GAIA or SimpleQA).**
>
> **This comment is incorrect.** As the web agent tasks (GAIA, SimpleQA) you mentioned is **a totally different line of research** although they both share the same term “web agent” with our work.
>
> First, we want to clarify, for the term “web agent” in recent research works, it can refer to two different lines of research:
>
> 1. **Search-based web agent** (deep research agent) [1-3]. Research works such as WebDancer and SearchR1 introduce agents for solving information-heavy questions, including GAIA, BrowserComp, and HLE. The action space of these agents usually consists of search-related tools, such as Google Search, Wikipedia Search, and web-page search. Their observation space is limited to the query and the API-returned results.
> 2. **GUI-based web agent** [4-10]. These web agents simulate human interaction with web elements using actions such as “click,” “type,” and “scroll,” instead of search APIs. The observation space is also different, as these agents issue actions directly based on the current web page (HTML code, Accessibility Tree, or image). Most of these agents are formulated with a sole focus on web navigation (interacting with the web environment) and overlook the importance of information extraction and data analytics in GUI tasks. This is our key observation, and we decouple the different required capabilities that current GUI-based web agents have been blending together.
>
> **Given this, we kindly request the reviewer to revisit the comment and check whether the review is based on a misunderstanding or an incorrect premise regarding our research focus.**
>
> However, we still appreciate the reviewer’s advice and have included another GUI-based web agent benchmark, WebVoyager, and compared it with our best baseline:
>
> **[Table Q3]. Performance of WebDART and AgentOccam on WebVoyager. (GPT-4o backbone)**
>
> | Agents     | Accuracy |
> | ---------- | -------- |
> | AgentOccam | 41.1     |
> | WebDART    | 45.7     |
>
> The main reason we did not include WebVoyager as one of our benchmarks is that the environment of this benchmark is not built on a static web environment like the WebArena family of benchmarks, which leads to changing task labels over time and low reproducibility. For example, the ground truth of one task sample in WebVoyager is “Vegan Chocolate Chip, Oatmeal, and Nut Cookies, 4.9 stars, 67 viewers” from the AllRecipes website, but now the ground truth has changed to “Vegan Chocolate Chip, Oatmeal, and Nut Cookies, 4.9 stars, 69 viewers.” Following AgentOccam, we chose a subset of the benchmark that does not require human evaluation and corrected the labels manually. Empirically, we find that WebDART still outperforms the best baseline, AgentOccam, which demonstrates the generalizability of WebDART across various benchmarks.

---

> ### Author Response · Authors · 2025-11-25
> **Response to Reviewer bz3S (3/3)**
>
> **[Q4]** **Most baseline performances are cited from prior studies, resulting in comparisons primarily against older methods.**
>
> First, given our clarification in Q3, it is neither reasonable nor practical for us to compare WebDART with any recent search-based web agent baselines, as they belong to a different research field.
>
> Secondly, the selection of baseline methods in our experiments is not based on time, but on the strongest open-sourced agents in the WebArena leaderboard. At the time of our submission, among the **top-performing open-sourced agents in WebArena**, only AgentSymbiotic and Learn-by-Interact were not included, and the reason was already specified in our paper Appendix A.2.1: as two data-driven methods, their performance depends heavily on their proprietary retrieval-augmented generation (RAG) databases. Because neither of these works has released their databases, a direct comparison would not be fair or reproducible, and we therefore exclude them from our evaluation.
>
> We believe that **the selection of baselines should be determined by performance** and should always compare with the SOTA method. Therefore, we disagree with your comment that “most baseline performances are cited from prior studies, resulting in comparisons primarily against older methods.”
>
> Lastly, the referred WebWalker is not comparable, as WebWalker focuses on a different task setting similar to search-based web agents—WebQA. Although, unlike search-based web agents, WebWalker does not utilize search APIs as tools, it is constrained to issuing only “click” actions to navigate between web pages and cannot complete even simple navigation tasks in WebArena, as WebArena requires handling a much more complex action space.
>
> Reference:
>
> [1] Hu, Mengkang, et al. "Owl: Optimized workforce learning for general multi-agent assistance in real-world task automation." arXiv preprint arXiv:2505.23885 (2025).
>
> [2] Li, Kuan, et al. "WebSailor: Navigating Super-human Reasoning for Web Agent." arXiv preprint arXiv:2507.02592 (2025).
>
> [3] Ye, Rui, et al. "AgentFold: Long-Horizon Web Agents with Proactive Context Management." arXiv preprint arXiv:2510.24699 (2025).
>
> [4] Wei, Zhepei, et al. "Webagent-r1: Training web agents via end-to-end multi-turn reinforcement learning." arXiv preprint arXiv:2505.16421 (2025).
>
> [5] Marreed, Sami, et al. "Towards enterprise-ready computer using generalist agent." arXiv preprint arXiv:2503.01861 (2025).
>
> [6] Zhang, Ruichen, et al. "Symbiotic cooperation for web agents: Harnessing complementary strengths of large and small llms." arXiv preprint arXiv:2502.07942 (2025).
>
> [7] Su, Hongjin, et al. "Learn-by-interact: A data-centric framework for self-adaptive agents in realistic environments." arXiv preprint arXiv:2501.10893 (2025).
>
> [8] Yang, Ke, et al. "Agentoccam: A simple yet strong baseline for llm-based web agents." arXiv preprint arXiv:2410.13825 (2024).
>
> [9] Qi, Zehan, et al. "Webrl: Training llm web agents via self-evolving online curriculum reinforcement learning." arXiv preprint arXiv:2411.02337 (2024).
>
> [10] Zhang, Yao, et al. "Webpilot: A versatile and autonomous multi-agent system for web task execution with strategic exploration." *Proceedings of the AAAI Conference on Artificial Intelligence*. Vol. 39. No. 22. 2025.

---

> ### Author Response · Authors · 2025-11-26
>
> Dear Reviewer bz3S,
>
> We thank you for the time on reviewing and the constructive feedback again. We really hope to discuss further with you to see if our response answers your questions.
>
> We genuinely hope reviewer bz3S could kindly check our response. Thank you very much!

---

### Official Review · Reviewer_6YZq · 2025-11-02

**Soundness:** 2
**Presentation:** 2
**Contribution:** 1
**Rating:** 2
**Confidence:** 3

**Summary:**

This paper introduces WebDART, a training-free framework for LLM-based web agents that improves performance on long-horizon, multi-step web tasks. The method dynamically decomposes complex objectives into three subtasks: navigation, information extraction, and execution. It also allows continuously re-planning for these subtasks as new webpage elements appear.
WebDART aims to reduce the overload by letting a single frozen LLM focus on one sub-capability at a time and to improve sample efficiency by adapting plans on the fly. Results on WebArena and WebChoreArena show empirical effectiveness.

**Strengths:**

1. Paper is well-written and easy to understand.
2. Strong quantitative results: The WebDART framework seems to be quite effective and achieves consistent improvements across three model backbones and different web domains.

**Weaknesses:**

Major:

1. Limited novelty: The three core modules used in WebDART, i.e., navigation, extraction, and execution, have been widely used as standard prompting paradigms and is commonly seen in recent web agent works. The design is also quite heuristic and is only supported by intuition rather than systematic error analysis of previous work.
2. Lack of learning or adaptation: All components in WebDART are purely prompt-engineered and rule-based. There is no learning or self-improvement involved in the method to adapt the policy itself  and learn a truly intelligent agent. This limits the framework’s scalability and robustness when deployed beyond the benchmark environments.
3. Baseline selection: The baseline comparison focuses on earlier methods, e.g., Table 3 misses many recent baselines on the WebArena  leaderboard with much stronger results, so the credibility of many claims on empirical effectiveness needs to be questioned.
4. Missing related work on multi-agent web navigation systems.

Minor:
1. Missing citation on line 111.

**Questions:**

NA

---

> ### Author Response · Authors · 2025-11-25
> **Response to Reviewer 6YZq (1/3)**
>
> We thank Reviewer 6YZq for the valuable feedback. Please see our response below.
>
> **[Q1]** **Navigation, extraction, and execution is commonly seen in recent web agent works.**
>
> We politely disagree with this claim and provide our clarification for this question from the two aspects: 1) Factually inaccuracies: navigation, extraction, and execution are **not common** **for GUI-based web agents**; 2) Our main contribution is the **dynamic decomposition mechanism** that improves both agent performance and efficiency.
>
> 1. Navigation is usually **the only module** for GUI-based web agents. First, we want to clarify that the term “web agent” in recent research works can refer to two different lines of research.
>    - **Search-based web agents (deep research agents)** [3–5]: these agents are designed to solve information-intensive questions (e.g., GAIA[1], BrowserComp[2]) using search APIs. Some of these agents may involve information extraction (via web-page search tools) and execution (via calculation tools), but this is not the type of “web agent” we are focusing on.
>    - **GUI-based web agents** [6–12]: these agents simulate human interaction with web elements using actions such as “click,” “type,” and “scroll,” instead of relying on search APIs. The observation space is also different, as these agents issue actions directly based on the current web page (HTML code, Accessibility Tree, or image). We reviewed current SOTA methods on WebArena leaderboard, and found almost all of them focus only on the navigation module and do not involve task decomposition like WebDART.
>
> 2. In addition, proposing modules such as navigation, information extraction, and execution is not our main contribution. Our main insight is to decouple the sub-tasks that require different capabilities, rather than following traditional GUI-based web agents that couple different capabilities solely within navigation.
>
>    Although some GUI-based web agents may include a decomposition module (e.g., WebPilot includes a planner agent that decomposes the original task into several sub-tasks), such decomposition is fundamentally different from ours. Their decomposition primarily focuses on breaking down a long-horizon task into several short-horizon tasks that require the same capability. For example, a task such as “Create a ‘NolanFans’ repo, listing Nolan's Oscar-winning films in a README file” can be divided into three sub-tasks — “Go to posts discussing Nolan's Oscar-winning films,” “Create a ‘NolanFans’ repo in GitLab,” and “Create a README file listing Nolan's Oscar-winning films.” All of these sub-tasks are navigation tasks, just like the original one. In contrast, our method focuses on decomposing the task at the capability level, not at the task-horizon level.
>
>    Another main contribution is our solution for how to more reasonably decompose a task. We observed that when decomposing a complex task into sub-tasks with different capabilities, determining the proper granularity/responsibility of each sub-task can be challenging, because:
>
>    - decomposition at the beginning lacks prior knowledge about the environment (e.g., whether there is a usable shortcut to make navigation easier); and
>    - backbone LLMs' capabilities vary. They are usually strong at data analytics but weak at web navigation.
>
> Based on these observations, we propose a dynamic decomposition mechanism, which leads to improved efficiency and performance on both WebArena and WebChoreArena benchmarks.

---

> ### Author Response · Authors · 2025-11-25
> **Response to Reviewer 6YZq (2/3)**
>
> **[Q2] There is no learning or self-improvement involved in the method.**
>
> 1. Our method is a general web agent solution and can be seamlessly integrated with different adaptation algorithms. We did not include this because it is not the focus of our work, and if our method were to leverage experience across evaluation test cases during inference time, it would be unfair to compare it with other methods. In addition, all of the baselines compared in our experiments are not learning-based methods and are not trained. Therefore, it would be unfair to include training in our agent while comparing it with other methods.
> 2. One of our baselines, Agent Workflow Memory, is indeed a method with self-improvement. It summarizes common action sequence patterns and experience from previous successful trials. In our implementation, we first ran it on the full benchmark using GPT-5 and summarized the experience from all successful trials. Based on these experiences, AWM then evaluated the benchmark again, and we include this as its final performance.
>
> ​	The results are given in the table below. On both WebArena and WebChoreArena, our method outperformed the AWM.
>
> **[Table Q2]. Performance of WebDART and AWM (self-improvement method) on WebArena and WebChoreArena. (GPT-4o backbone)**
>
> |                   | Shopping | Admin | Reddit | Gitlab | Avg  |
> | ----------------- | -------- | ----- | ------ | ------ | ---- |
> | **WebArena**      |          |       |        |        |      |
> | AWM               | 32.1     | 29.1  | 54.7   | 35.0   | 37.7 |
> | WebDART           | 36.0     | 41.2  | 67.9   | 47.2   | 48.1 |
> | **WebChoreArena** |          |       |        |        |      |
> | AWM               | 3.4      | 8.8   | 4.5    | 4.7    | 5.4  |
> | WebDART           | 18.8     | 19.8  | 12.9   | 9.4    | 15.2 |
>
> **[Q3] Misses many recent baselines on the WebArena leaderboard.**
>
> In our experiments, we mainly focused on comparing with **the open-sourced agents**.
>
> 1. We did not compare with the agents Claude Code + GBOX MCP and IBM CUGA for two main reasons:
>
>    - **These agents were released in October 2025.** Although the preprint of IBM CUGA was available in February, its code was released only last month. Since they are considered concurrent work, it would have been difficult for us to compare with them during our original submission.
>
>    - **Both of these agents use their pre-defined MCP tools to improve agent performance.** This results in a major difference in action space compared to our agent and all of our baselines, creating an unfair advantage. In addition, Claude Code + GBOX MCP requires using Claude Code as the backbone model, which further limits its applicability to our experimental setting. This results in a major difference in action space compared to our agent and all of our baselines, creating an unfair advantage. In addition, Claude Code + GBOX MCP requires using Claude Code as the backbone model, which further limits its applicability to our experimental setting.
>
> 2. The comparison with AgentSymbiotic and Learn-by-Interact is also excluded, and the reason for this is stated in our paper Appendix A.2.1: as two data-driven methods, the performance of these methods depends heavily on their proprietary retrieval-augmented generation (RAG) databases. Because neither of these works has released their databases, a direct comparison would not be fair or reproducible, and we therefore exclude them from our evaluation.

---

> ### Author Response · Authors · 2025-11-25
> **Response to Reviewer 6YZq (3/3)**
>
> **[Q4] Missing related work on multi-agent web navigation systems.**
>
> Thank you for the suggestion. We agree that comparing WebDART with existing multi-agent frameworks offers valuable insight.
>  In our comparison, we include **WebPilot**, a representative multi-agent approach. WebPilot separates high-level reasoning from low-level execution through: (1) **a planner model** that produces step-by-step intentions, and (2) **an executor model** that translates these intentions into concrete browser actions. It also uses (3) **a self-correction module** to identify mismatches between the expected and actual webpage states and perform recovery.
>
> Because WebPilot does not release an open-source implementation, we reproduced it for evaluation on WebChoreArena. Please note that several hyper-parameters (e.g., reward weights, scroll budget, and certain MCTS-related parameters) were not specified in the original paper, so we assigned reasonable values—using reward weights of [0.5, 0.5], a scroll budget of 5, and standard defaults for the missing MCTS components, such as normalizing rewards to [0,1]. The experimental results using GPT-4o as the backbone model are reported below.
>
> **[Table Q4]. Comparison between WebPilot (multi-agent) and WebDART (GPT-4o backbone)**
>
> |               | Shopping | Admin | Reddit | Gitlab | Avg  |
> | ------------- | -------- | ----- | ------ | ------ | ---- |
> | WebChoreArena |          |       |        |        |      |
> | SteP          | 2.6      | 0.0   | 0.0    | 4.7    | 1.8  |
> | BrowserGym    | 0.9      | 5.5   | 2.3    | 3.9    | 3.2  |
> | AWM           | 3.4      | 8.8   | 4.5    | 4.7    | 5.4  |
> | AgentOccam    | 10.3     | 9.9   | 4.5    | 7.1    | 8.0  |
> | **WebPilot**  | 6.8      | 2.2   | 4.5    | 3.9    | 4.4  |
> | **WebDART**   | 18.8     | 19.8  | 12.9   | 9.4    | 15.2 |
>
> We highlight that **WebDART outperforms the multi-agent baseline and achieves the highest performance in every domain**. This further verifies the effectiveness of our method.
>
> Reference:
>
> [1] Mialon, Grégoire, et al. "Gaia: a benchmark for general ai assistants." The Twelfth International Conference on Learning Representations. 2023.
>
> [2] Wei, Jason, et al. "Browsecomp: A simple yet challenging benchmark for browsing agents." arXiv preprint arXiv:2504.12516 (2025).
>
> [3] Hu, Mengkang, et al. "Owl: Optimized workforce learning for general multi-agent assistance in real-world task automation." arXiv preprint arXiv:2505.23885 (2025).
>
> [4] Li, Kuan, et al. "WebSailor: Navigating Super-human Reasoning for Web Agent." arXiv preprint arXiv:2507.02592 (2025).
>
> [5] Ye, Rui, et al. "AgentFold: Long-Horizon Web Agents with Proactive Context Management." arXiv preprint arXiv:2510.24699 (2025).
>
> [6] Wei, Zhepei, et al. "Webagent-r1: Training web agents via end-to-end multi-turn reinforcement learning." arXiv preprint arXiv:2505.16421 (2025).
>
> [7] Marreed, Sami, et al. "Towards enterprise-ready computer using generalist agent." arXiv preprint arXiv:2503.01861 (2025).
>
> [8] Zhang, Ruichen, et al. "Symbiotic cooperation for web agents: Harnessing complementary strengths of large and small llms." arXiv preprint arXiv:2502.07942 (2025).
>
> [9] Su, Hongjin, et al. "Learn-by-interact: A data-centric framework for self-adaptive agents in realistic environments." arXiv preprint arXiv:2501.10893 (2025).
>
> [10] Yang, Ke, et al. "Agentoccam: A simple yet strong baseline for llm-based web agents." arXiv preprint arXiv:2410.13825 (2024).
>
> [11] Qi, Zehan, et al. "Webrl: Training llm web agents via self-evolving online curriculum reinforcement learning." arXiv preprint arXiv:2411.02337 (2024).
>
> [12] Zhang, Yao, et al. "Webpilot: A versatile and autonomous multi-agent system for web task execution with strategic exploration." *Proceedings of the AAAI Conference on Artificial Intelligence*. Vol. 39. No. 22. 2025.

---

> ### Author Response · Authors · 2025-11-26
>
> Dear Reviewer 6YZq,
>
> We thank you for the time on reviewing and the constructive feedback again. We really hope to discuss further with you to see if our response answers your questions.
>
> We genuinely hope reviewer 6YZq could kindly check our response. Thank you very much!

---

### Note · Authors · 2026-01-06

I have read and agree with the venue's withdrawal policy on behalf of myself and my co-authors.